# Genetic or pharmacological GHSR blockade has sexually dimorphic effects in rodents on a high-fat diet
András H. Lékó[1,2,19], Adriana Gregory-Flores[1,3,19], Renata C. N. Marchette [3,19], Juan L. Gomez[4], Janaina C. M. Vendruscolo[3], Vez Repunte-Canonigo[5], Vicky Choung [1,3], Sara L. Deschaine[1], Kimberly E. Whiting[1,3], Shelley N. Jackson[6], Maria Paula Cornejo[7], Mario Perello[7], Zhi-Bing You[8], Michael Eckhaus[9], Karuna Rasineni[10,11,12], Kim D. Janda [13], Barry Zorman[14], Pavel Sumazin [14], George F. Koob[3], Michael Michaelides [4], Pietro P. Sanna [5], Leandro F. Vendruscolo [15,20] ✉ & Lorenzo Leggio [1,6,16,17,18,20] ✉

The stomach-derived hormone ghrelin regulates essential physiological functions. The ghrelin receptor (GHSR) has ligand-independent actions; therefore, *GHSR* gene deletion may be a reasonable approach to investigate the role of this system in feeding behaviors and diet-induced obesity (DIO). Here, we investigate the effects of a long-term (12-month) high-fat (HFD) versus regular diet on obesity-related measures in global GHSR-KO and wild-type (WT) Wistar male and female rats. Our main findings are that the *GHSR* gene deletion protects against DIO and decreases food intake during HFD in male but not in female rats. *GHSR* gene deletion increases thermogenesis and brain glucose uptake in male rats and modifies the effects of HFD on brain glucose metabolism in a sex-specific manner, as assessed with small animal positron emission tomography. We use RNA-sequencing to show that GHSR-KO rats have upregulated expression of genes responsible for fat oxidation in brown adipose tissue. Central administration of a novel GHSR inverse agonist, PF-5190457, attenuates ghrelin-induced food intake, but only in male, not in female mice. HFD-induced binge-like eating is reduced by inverse agonism in both sexes. Our results support GHSR as a promising target for new pharmacotherapies for obesity.

Obesity is a chronic disease that leads to serious health consequences, decreased life expectancy, and significant health care costs. In 2017-2018, obesity affected over 42% of adults in the United States[1]. There is a critical need to investigate novel pathways related to obesity and guide new treatment approaches[2].

Ghrelin is an acylated 28 amino acid orexigenic hormone synthesized by endocrine cells, primarily located in the stomach, implicated in metabolism, reward, and food intake[3,4]. The *GHRL* gene encodes the posttranslational formation of ghrelin by cleaving preproghrelin into proghrelin to be acylated by the ghrelin O-acyltransferase (GOAT)[5–7]. Ghrelin acts on the growth hormone secretagogue receptor (GHSR), which is widely expressed both in the central nervous system and periphery, including the stomach, adrenal glands, and adipose tissue[8–10]. Recently, LEAP2 was identified as an endogenous GHSR antagonist that inhibits ghrelin's effects and blocks constitutive GHSR activity[11].

Circulating ghrelin levels fluctuate along with feeding states, increasing during fasting and decreasing during satiety. Both peripheral and central administration of ghrelin stimulates appetite, food seeking, and food consumption by linking peripheral signaling of the hormone to central modulation of feeding behaviors[12]. However, whereas ghrelin-related mechanisms in appetite, food intake, and reward have been established, this information has failed to translate into novel effective treatments for obesity, suggesting the need for further investigation on the role of ghrelin in obesity.

High-fat diets (HFDs), that range between 30 and 78% of total caloric content, can induce obesity in both rodents and humans[13–15]. Relevant to our study, food intake reduces postprandial plasma ghrelin levels, an effect that is blunted in obese people[16,17]. Not only is this dysregulation affecting feeding behaviors, but ghrelin also induces adiposity under HFDs[18].

The orexigenic and obesity-promoting effects of ghrelin are regulated through the GHSR[19], which is expressed widely in the body, including in the

hypothalamus, ventral tegmental area[10,20], and adipose tissue[21]. Interestingly, knockout rodents lacking the ghrelin peptide show little to no behavioral, physiological, or metabolic changes compared with wild-type (WT) controls[22,23]. These apparent discrepancies may be explained, at least in part, by some ligand-independent actions of GHSR including its high intrinsic activity, even with the lack of ghrelin binding[24,25]. Therefore, the deletion of *GHSR* gene provides a compelling proof-of-concept approach to investigate all the roles of this receptor in feeding behaviors and obesity. Toward this end, in the present work, we investigated HFD *versus* regular diet in GHSR-KO male and female rats. These are CRISPR/Cas9-based global GHSR-KO Wistar rats, similar to that presented a phenotype of slightly reduced body weight and decreased food consumption under normal feeding (e.g., non-HFD) conditions, as *Zallar* et al. described earlier[26]. Those GHSR-KO male rats were also insensitive to ghrelin's growth hormone (GH) secreting and orexigenic effects after systemic injection of ghrelin and had an increased percentage of brown adipose tissue (BAT) in total body weight[26].

Although interesting and promising, the experiments described above[26] were limited to male rats. Expanding this work to both sexes is therefore important, even more so, given that ghrelin seems to play a role in sex-related differences in the physiology of eating[27]. Furthermore, in the previous work[26], the rats were only fed a regular diet, and their body weight was followed and measured only throughout 13 weeks in adulthood. To further study GHSR in a condition that better resembles what is observed in humans who have prolonged exposure to high-calorie food and develop obesity, we investigated the role of GHSR, in GHSR-KO and WT rats of both sexes, that were exposed to an HFD for 12 months. We used a comprehensive approach that involved metabolic, behavioral, endocrine, neuroimaging, and molecular measurements. We hypothesized that GHSR-KO male and female rats would be more resistant to HFD-induced behavioral and physiological changes compared with WT rats. Hence, they would also be more resistant to the HFD-led diet-induced obesity (DIO) phenotype.

HFD also contributes to DIO through inducing binge eating and hyperphagia[28]. GHSR constitutive activity in dopaminergic neurons is linked to hyperphagia induced by palatable foods, like HFD[29,30]. In the current study, in male and female mice, we administered the first GHSR inverse agonist, PF-5190457, which progressed to clinical development, and was found well-tolerable by oral administration in humans[31,32]. Furthermore, PF-5190457 reduced cue-induced food craving and food-seeking behavior in humans[32]. We hypothesized that, first, PF-5190457 would prove its pharmacodynamic efficacy by diminishing ghrelin-induced food intake, and second, it would reduce HFD-induced hyperphagia. This pharmacological intervention may strengthen the translational potential of GHSR as a target against DIO and binge eating.

## Results

### Body Weight and Food Consumption
We measured body weight (BW, expressed in g) and food intake (expressed in g) twice weekly for 12 months, from the age of 2-4 months, to assess the effect of genotype on DIO. Rats were fed either a regular chow diet (Chow) with 24% protein, 58% carbohydrate, 18% fat (2.89 kcal/g) or an HFD with 20% protein, 20% carbohydrate, 60% fat (5.24 kcal/g).

For BW in males, with a 3-way repeated-measures (RM) ANOVA, we found significant effects of Time ($F_{1.32, 31.65} = 377.72$, $p < 0.0001$), Genotype ($F_{1, 24} = 7.66$, $p = 0.011$; WT > GHSR-KO) and Diet ($F_{1, 24} = 19.39$, $p = 0.0002$; HFD > chow), and a Time x Diet interaction ($F_{11, 264} = 16.58$, $p < 0.0001$). *Post hoc* comparisons based on the Time x Diet interaction indicated that HFD-fed male rats weighted significantly more than chow-fed male rats from month 2 until the end of the study ($p < 0.05$; Fig. 1a). When analyzing HFD and chow-fed groups separately with 2-way RM ANOVA, the effect of Genotype was only seen in the HFD-fed animals, i.e., GHSR-KO rats' weight was significantly lower compared to WT ($p = 0.017$; Fig. 1a).

For weekly food consumption in HFD-fed males, the 2-way RM ANOVA showed significant effects of Time ($F_{1.934, 25.14} = 6.609$, $p = 0.005$)

and Genotype ($F_{1, 13} = 10.81$, $p = 0.006$), indicating that GHSR-KO rats on the HFD consumed less food compared to WT group (Fig. 1b).

For BW in female rats, 3-way RM ANOVA showed significant effects of Time ($F_{1.59, 46.04} = 298.073$, $p < 0.001$), Diet ($F_{1, 29} = 6.734$, $p < 0.001$), and a Time x Diet interaction ($F_{1.59, 46.04} = 12.947$, $p < 0.001$). *Post hoc* comparisons indicated that HFD-fed female rats weighted significantly more than chow-fed female rats in months 7-12 ($p < 0.05$; Fig. 1c), regardless of the genotype. Separate 2-way RM ANOVAs in different diet groups did not show any significant effect of Genotype in females (Fig. 1c).

For weekly food consumption in HFD-fed female rats, only a significant effect of Time ($F_{2.653, 37.14} = 3.986$, $p = 0.018$) was detected by the 2-way RM ANOVA (Fig. 1d).

We also calculated the weekly feed efficiency (weight gain [mg]/energy consumed [kcal]) and did not find any statistically significant difference between HFD-fed male and female rats (Supplementary Results and Supplementary Fig. S1).

The effects described above were unlikely due to differences in locomotion or anxiety-like behavior, as there were no genotype differences in an open field test or a novelty-suppressed feeding test (Supplementary Results, Supplementary Figs. S2,3, and Supplementary References).

### Carcass analysis
White adipose tissue (WAT), including visceral fat pads (gonadal, inguinal), intrascapular BAT, brain, liver, spleen, and adrenal glands, were collected and weighed after decapitation under deep isoflurane anesthesia. We analyzed the tissue weights normalized to the body weight and expressed them as % of total body weight.

In males, but not females we found a significant effect of Genotype ($F_{1, 24} = 6.489$, $p = 0.018$; GHSR-KO < WT) for the percent of gonadal WAT weight (Supplementary Fig. S4c). We detected a significant effect of Diet on relative WAT weight, as HFD increased the percent of gonadal WAT weight in males and females (Supplementary Fig. S4c, d), and the percent of inguinal WAT weight in males (Supplementary Fig. S4e). A 2-way ANOVA showed an effect of Diet on the percentage of liver weight, indicating decreased liver weights in HFD-fed male and female rats compared with chow-fed rats, regardless of Genotype (Supplementary Table 1). Given these results on liver weights, we further investigated hepatic lipid accumulation via qualitative histological analysis of hematoxylin-eosin-stained liver slices. Livers of HFD-fed male and female rats showed mild-to-severe steatosis, whereas livers of chow-fed rats in both sexes showed either no sign or only mild-to-moderate steatosis (Supplementary Fig. S5). We also quantified the hepatic steatosis/fat accumulation by measuring the concentration of hepatic triglycerides. HFD caused higher triglyceride levels in both males and females, whereas Genotype did not have a significant effect. The analysis also revealed that males have increased concentration of triglycerides compared to females, but this effect of sex was only significant in the HFD animals (Supplementary Fig. S6). For details see Supplementary Results.

### Small animal Positron Emission Tomography (PET) – Whole Brain
A small animal PET scan with [$^{18}$F] fluorodeoxyglucose (FDG) was used to measure glucose uptake in the whole brain, 50 weeks after initiation of Chow or HFD. First, the main effect of Diet was analyzed separately in different sexes and genotypes. In male rats, HFD led to a decrease in FDG uptake, regardless of genotype. WT males were more affected by the diet, than GHSR-KO males (Fig. 2a). In females, HFD caused minor decreases in FDG uptake in WT rats, but marked decreases were detected in GHSR-KO rats. HFD led also to increases in multiple areas, distinct from decreases, in both WT and GHSR-KO females, and overall, GHSR-KO females were more affected by HFD (Fig. 2b).

Second, the main effect of Genotype was analyzed separately in males and females. *GHSR* deletion caused increased FDG uptake throughout the brain, regardless of Diet, and this effect was stronger in males (Fig. 3). In females, GHSR deletion also led to increased uptake in the majority of the brain areas; however, we detected a decrease in multiple regions.

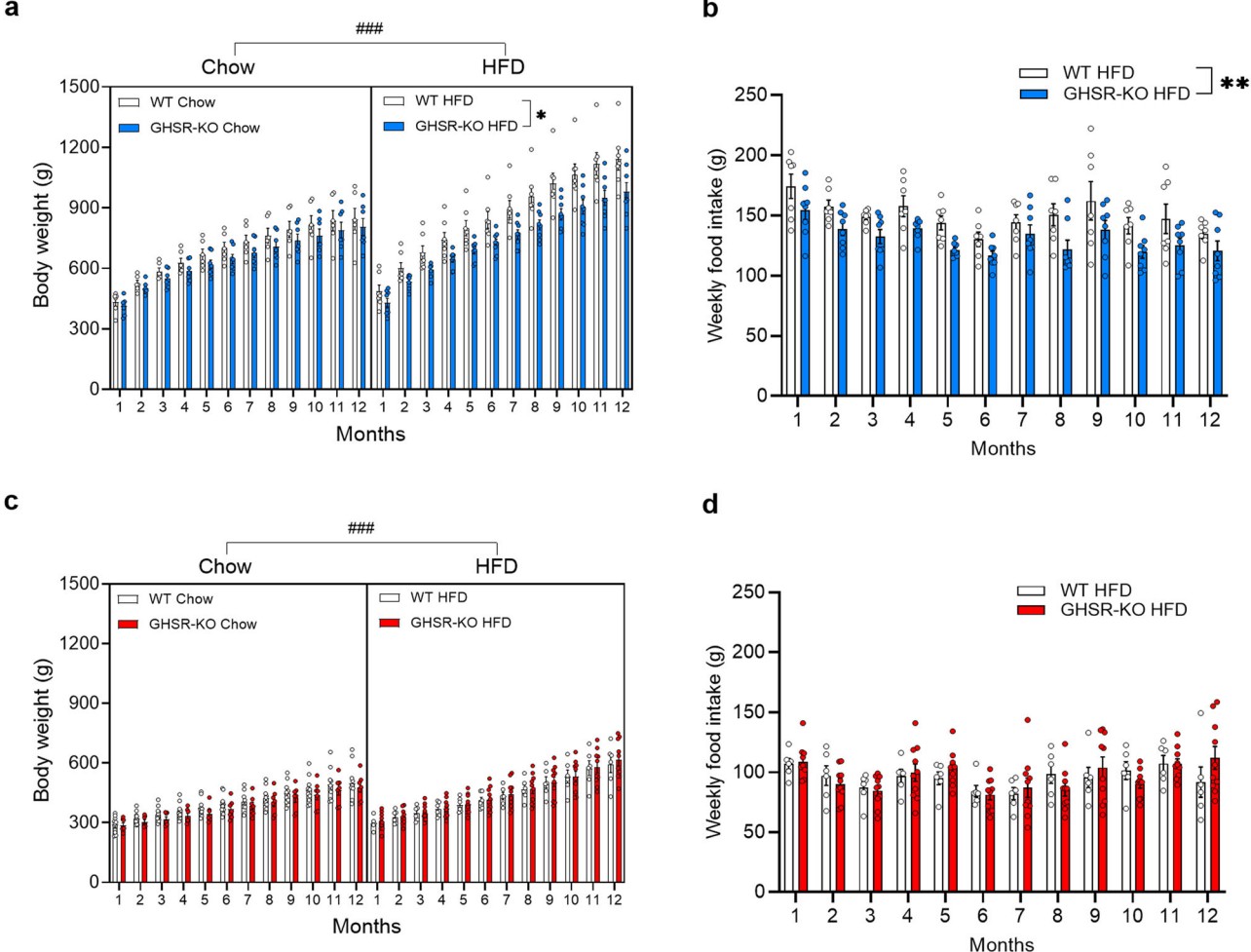

**Fig. 1 | GHSR deletion has a protective effect against diet-induced obesity and decreases food intake in male, but not female rats exposed to an HFD. a** GHSR-knockout (KO) male rats that had *ad libitum* access to a high-fat diet (HFD) weighed significantly less than wild-type (WT) male rats (*$p < 0.05$). Male rats that were maintained on an HFD weighed significantly more than male rats that were maintained on a chow diet from months 2-12 (###$p < 0.001$). WT chow: $n = 6$. GHSR-KO chow: $n = 7$. WT HFD: $n = 7$. GHSR-KO HFD: $n = 8$. **b** GHSR-KO male rats on the HFD consumed significantly less food weekly than WT rats on the HFD

(**$p < 0.01$, difference between WT and GHSR-KO rats regardless of time). **c** Regardless of genotype, female rats that were maintained on an HFD weighed significantly more than female rats that were maintained on a chow diet from months 7–12 (###$p < 0.001$). WT chow: $n = 10$. GHSR-KO chow: $n = 7$. WT HFD: $n = 6$. GHSR-KO HFD: $n = 10$. **d** There was no difference between the weekly food intake of WT and GHSR-KO female rats on HFD. Data are expressed as mean ± SEM.

For a more detailed analysis of FDG uptake by different brain areas, see Supplementary Results and Supplementary Fig. S7.

### Brown Adipose Tissue (BAT) Thermogenesis
To assess BAT thermogenesis, we determined the surface body temperature of the interscapular region, where BAT is in rats. In male rats, a 2-way ANOVA indicated a significant main effect of Genotype ($F_{1, 24} = 8.674$, $p = 0.007$), with GHSR-KO male rats having higher temperatures than WT male rats, regardless of the Diet (Fig. 4a). In females, we did not detect significant effects for the temperature of the interscapular region (Fig. 4b).

### Endocrine assays
We did not detect any effect of Genotype in the glucose tolerance test (Supplementary Fig. S8). In the insulin tolerance test, GHSR-KO males on HFD had slightly, but significantly, higher glucose levels during the test than WT males on the same diet (Supplementary Fig. S9). In the terminal hormone analysis, GHSR-KO chow-fed male rats had higher progesterone levels compared to the WT group with the same diet, and HFD-fed group with the KO genotype (Supplementary Fig. S10h). Aldosterone levels were

increased in the chow-fed group (Supplementary Fig. S10i). Leptin was higher with HFD, regardless of genotype (Supplementary Fig. S10j). As an exploratory analysis, we investigated sex differences among the hormones directly related to the ghrelin system and detected significantly higher LEAP2 levels in males, regardless of diet and genotype, compared with females (Supplementary Fig. S11). For details, see Supplementary Information.

### RNA Sequencing (RNA-Seq) and lipidomic analyses of the adipose tissue in males
Gene expression was profiled by RNA-Seq in males because the effect of GHSR-KO was more prominent compared to females in the experiments described above. We used Gene Set Enrichment Analysis (GSEA) for pathway analysis[33] in conjunction with genesets from the Molecular Signatures Database (MSigDB). Comparison of gene expression in the BAT of GHSR-KO *vs.* WT on the HFD by GSEA showed an upregulation of genesets in GHSR-KO that were representative of the tricarboxylic acid (TCA) cycle, oxidative phosphorylation, electron transfer, fat metabolism, β-oxidation, mitochondrial genes, glycolysis genes, and skeletal muscle-related genes (Fig. 5a–c and Supplementary Data 2); these findings

**Fig. 2 | Small animal PET; effect of Diet.** Statistical parametric maps ($T = 1.72$; $p < 0.05$) of [$^{18}$F] fluorodeoxyglucose (FDG) uptake as a function of Genotype and Diet ($n = 6$/treatment group). **a** Males: higher FDG uptake in chow vs. HFD in WT (green) and GHSR-KO (blue). **b** Females: higher FDG uptake in chow vs. HFD in WT (green) and GHSR-KO (blue); lower FDG uptake in chow vs. HFD in GHSR-KO (red).

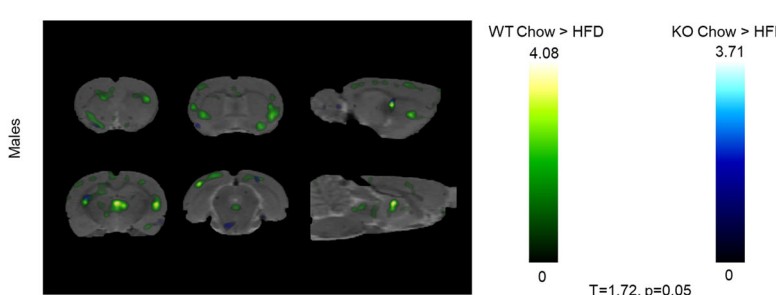

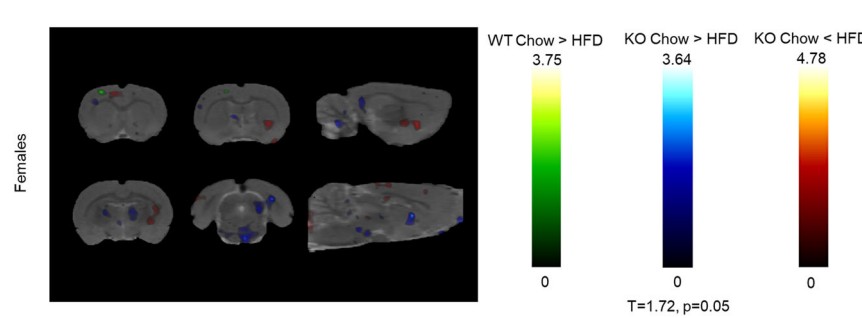

**Fig. 3 | Small animal PET; effect of Genotype.** Statistical parametric maps ($T = 1.72$; $p < 0.05$) of [$^{18}$F] fluorodeoxyglucose (FDG) uptake as a function of Genotype and Sex, regardless of Diet. Males (blue): higher FDG uptake in KO vs. WT ($n = 12$ each). Females (red): higher FDG uptake in KO vs. WT ($n = 12$ each).

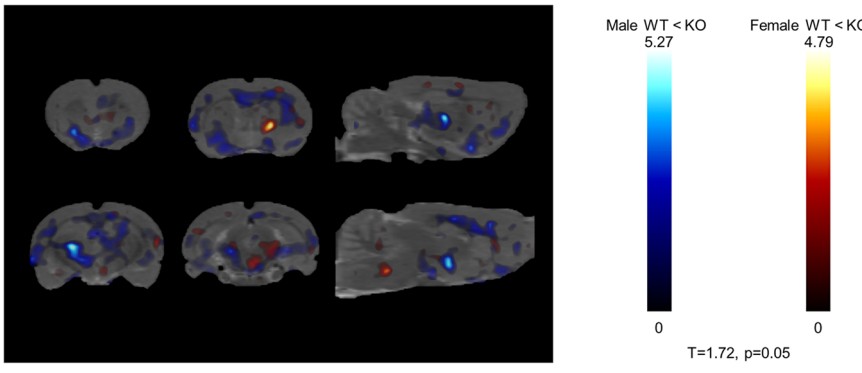

**Fig. 4 | In the brown adipose tissue (BAT), GHSR KO leads to higher interscapular temperature in male, but not in female, rats. a** GHSR-knockout (KO) male rats ($n = 7$ Chow, $n = 8$ HFD) had higher temperatures of the interscapular region compared with WT rats ($n = 6$ Chow, $n = 7$ HFD), regardless of the Diet (**$p < 0.01$). **b** There were no significant differences between WT ($n = 10$ Chow, $n = 6$ HFD) and GHSR-KO ($n = 7$ Chow, $n = 10$ HFD) female rats fed a chow diet or HFD on temperature of the interscapular region. Data are expressed as mean ± SEM. Circles represent each individual rat.

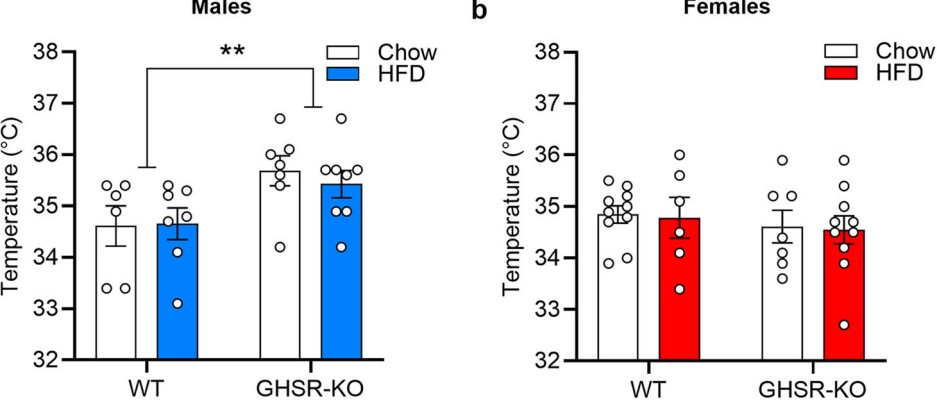

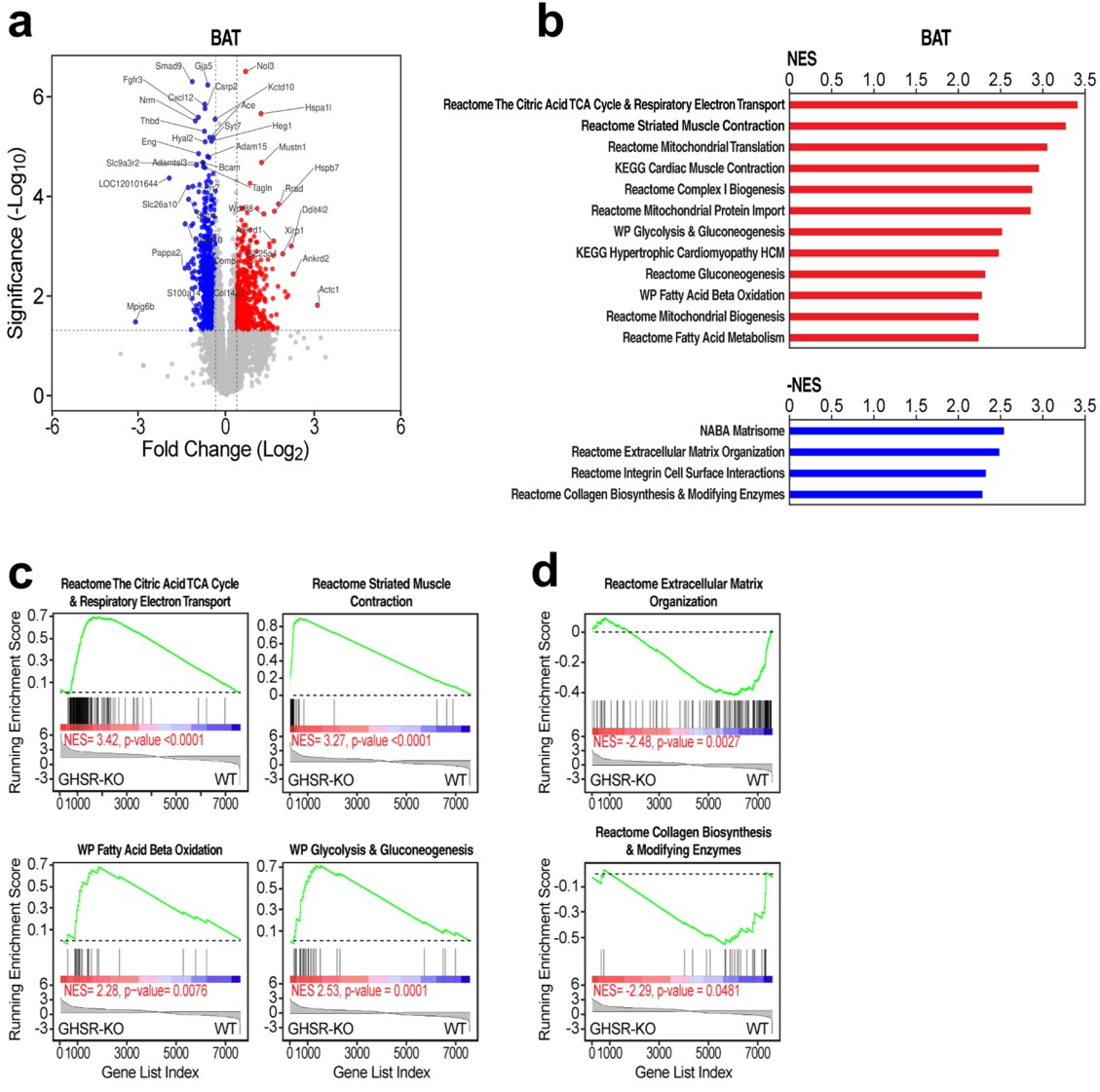

**Fig. 5 | GHSR deletion induced a specific oxidative and myogenic gene signature in BAT of male rats. a** Scatter (volcano) plot that depicts differentially expressed genes in the BAT of GHSR-KO rats on the HFD *vs.* WT rats on the HFD. X and Y axes, log2(Fold Change) and -log10 (*p*-value). **b** Gene Set Enrichment Analysis (GSEA) pathway analysis for BAT of GSHR-KO on the HFD vs. WT rats on the same diet. GSHR-KO shows higher expression of genesets involved in mitochondrial function and energy metabolism, including TCA cycle, oxidative phosphorylation, electron transfer, fat metabolism, β-oxidation, and skeletal muscle-related genes (red bar graphs). Conversely, WT rats on the HFD showed higher expression of extracellular matrix-related genes (blue bar graphs) (genesets shown have *FWER p*-values < 0.05*; overlapping genesets not shown; complete list is in Supplementary

Data 2). NES; normalized enrichment score[33]. **c** GSEA plots of representative gen-esets upregulated in the BAT of GHSR-KO on the HFD vs. WT BAT on the same diet, which indicate upregulation of TCA cycle and respiratory electron transport; fat and glucose metabolism genes; and increased expression of BAT skeletal muscle-like gene program. **d** GSEA plots of representative genesets that indicate upregulation of extracellular matrix and collagen genes in WT BAT on the HFD vs. GHSR-KO BAT on the on the same diet. Changes in the expression of the pathway in the GSEA plots, such as the ones in **c**, are indicated by the asymmetric distribution of genes in the geneset (vertical bars) and of the line plot of the running NES toward the side of the condition indicated underneath: GHSR-KO or WT rats.

were suggestive of increased fat oxidation. Conversely, the BAT of WT rats fed with the HFD showed greater expression of extracellular matrix, integrins and collagen biosynthesis, and modifying enzyme genes than the BAT of GHSR-KO rats on the same diet (Fig. 5a, b, d and Supplementary Data 2). The differential gene expression patterns of the BAT of GHSR-KO and WT were also evident in chow-fed rats (Supplementary Fig. S12a and Supplementary Data 3); however, they were more pronounced in HFD-fed rats (Supplementary Fig. S12b and Supplementary Data 4). Genesets related to TCA cycle, β-oxidation, mitochondrial genes, and skeletal muscle were also induced in WT rats by the HFD (Supplementary Fig. S12c and Supplementary Data 5), but to a lower extent than in GHSR-KO rats (Supplementary Fig. S12d and Supplementary Data 6).

Gene expression analysis of the WAT of male WT rats on the HFD vs. GHSR-KO on the same diet showed increased expression of inflammation-related pathways, whereas pathways linked to telomerase activity were up-regulated in WAT of GHSR-KO rats on the HFD (Fig. 6 and Supplementary Data 7).

The results of the *lipidomic analysis* of WAT and BAT by mass spectrometry show that *GHSR* gene deletion significantly increased a subset of triglyceride (TG) species in the WAT in males on a chow diet (Supplementary Fig. 13a and Supplementary Table 2). In BAT samples only two TG species were elevated in GHSR-KO males compared to WT (chow) (Supplementary Fig. 13a and Supplementary Table 3). Detailed results are described in Supplementary Information and Supplementary Fig.13, Supplementary Tables 2, 3.

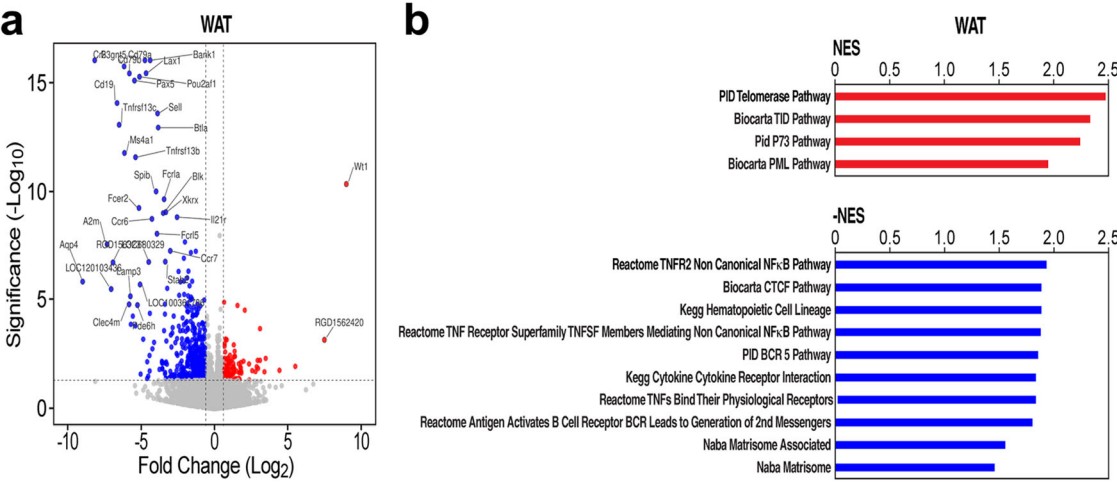

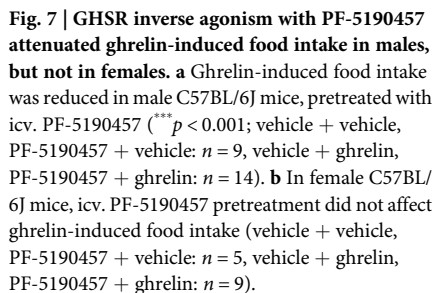

**Fig. 6 | GHSR deletion induced telomerase activity and prevented against inflammatory gene expression in WAT of male rats. a** Scatter (volcano) plot that depicts differentially expressed genes in the WAT of GHSR-KO rats on the HFD vs. WT rats on the same diet. X and Y axes, log2(Fold Change) and -log10(unadjusted p-value). X and Y values capped at 9 and 16, respectively. **b** Gene Set Enrichment Analysis (GSEA) pathway analysis for WAT of GHSR-KO vs. WT rats on the HFD suggests increased telomerase activity (red bar graphs) in GHSR-KO rats, while genesets with increased expression in WT indicate increased inflammation (blue bar graphs) (*NOM p-*values *< 0.05*). Complete lists of significantly differentially regulated genesets are in Supplementary Data 7–10. NES normalized enrichment score[33].

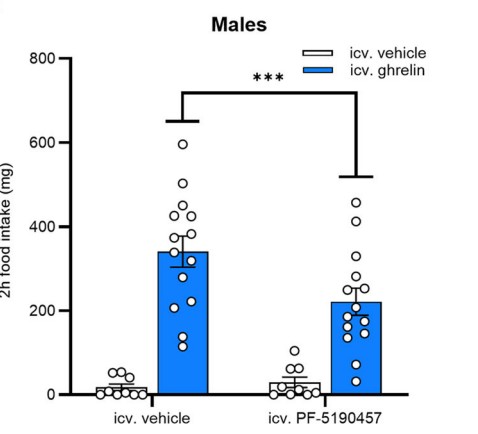

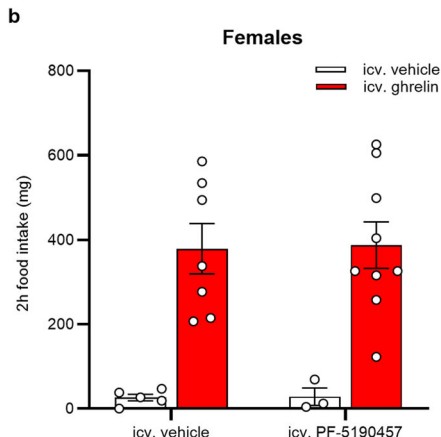

**Fig. 7 | GHSR inverse agonism with PF-5190457 attenuated ghrelin-induced food intake in males, but not in females. a** Ghrelin-induced food intake was reduced in male C57BL/6J mice, pretreated with icv. PF-5190457 (***p < 0.001*; vehicle + vehicle, PF-5190457 + vehicle: *n* = 9, vehicle + ghrelin, PF-5190457 + ghrelin: *n* = 14). **b** In female C57BL/6J mice, icv. PF-5190457 pretreatment did not affect ghrelin-induced food intake (vehicle + vehicle, PF-5190457 + vehicle: *n* = 5, vehicle + ghrelin, PF-5190457 + ghrelin: *n* = 9).

## Central GHSR inverse agonism with PF-5190457 in WT and GHSR-KO mice

Notably, the studies reported here unmasked a dramatic sexually dimorphic phenotype as the deletion of GHSR protected against obesity in male, but not female, HFD-fed rats. Thus, we hypothesized that PF-5190457, the first GHSR antagonist/inverse agonist that has progressed to clinical development and is well-tolerable by oral administration in humans[31,32], would also have sexually dimorphic effects in rodents.

First, we tested if intracerebroventricularly (ICV) administered PF-5190457 blocked ICV ghrelin-induced food intake in WT male and female C57BL/6J mice. We measured the amount of consumed chow pellets during 2 h after ICV vehicle/PF-5190457 plus ICV vehicle/ghrelin administration. In males, 2-way RM ANOVA indicated a significant ghrelin x PF-5190457 interaction ($F_{1, 20}$ = 11.67, *p* = 0.003), and *post hoc* comparisons indicated that PF-5190457 + ghrelin-treated mice consumed less chow than vehicle + ghrelin-treated mice (*p* < 0.001; Fig. 7a). In females, we did not see any ghrelin x PF-5190457 interaction (Fig. 7b). Ghrelin's main effect on food intake was significant in both sexes, regardless of the pretreatment (males: $F_{1, 22}$ = 41.72, *p* < 0.0001; females: $F_{1, 12}$ = 28.27, *p* < 0.001; Fig. 7). In addition, we quantified the number of c-Fos immunoreactive (IR) cells in the hypothalamic arcuate nucleus (Arc) after the 2 h food intake, in male and female mice

treated with ICV ghrelin and PF-5190457. GHSR inverse agonist pre-treatment results in lower number c-Fos-IR cells in ghrelin-treated male (*p* = 0.003) and female mice (*p* < 0.022) (Supplementary Fig. S14a, b). Interestingly, PF-5190475 did not affect overnight food intake either in male or in female mice, compared to vehicle-treated animals (Supplementary Fig. S14c, d).

Next, we tested the effect of ICV PF-5190457 on HFD-induced binge-like eating in satiated mice. We used a protocol with 2-hours HFD exposure, repeated throughout 4 consecutive days[29,30]. 2-way RM ANOVA showed a significant main effect of Time (males: $F_{3, 63}$ = 4.391; *p* = 0.007; females: $F_{1.86, 31.55}$ = 4.444; *p* = 0.022), meaning that repeated HFD-exposure induced binge-like eating. Main effect of Treatment (ICV PF-5190457 < vehicle) was also significant in both males ($F_{1, 23}$ = 20.86; *p* = 0.0001) and females ($F_{1, 17}$ = 10.18; *p* = 0.005), while Treatment x Time interaction was only significant in males ($F_{3, 63}$ = 12.97, *p* < 0.0001, Fig. 8a, b) meaning that PF-5190457-treated male mice consumed less HFD than vehicle-treated animals on days 2, 3 and 4 of the protocol (*p* < 0.0001). Finally, we exposed GHSR-KO male mice[34] to the same HFD binge-like eating protocol and analyzed the effect of PF-5190457 administration. As expected, 2-way RM ANOVA detected no differences in HFD intake between PF-5190457- and vehicle-treated GHSR-KO male mice (Fig. 8c), and also HFD did not induce binge-like eating since the main effect of Time was not significant.

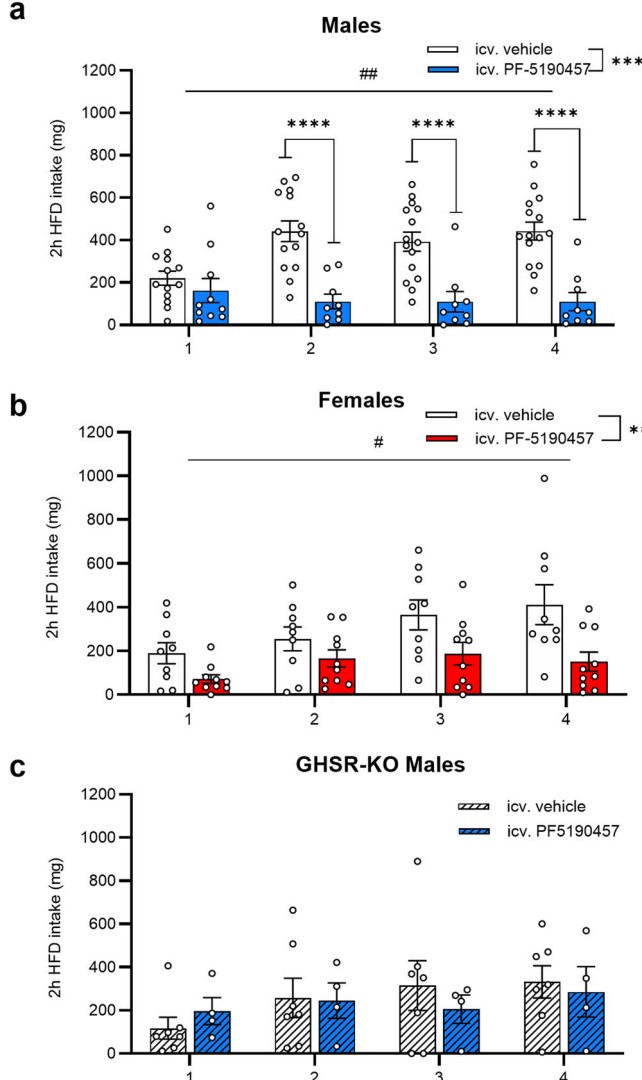

**Fig. 8 | GHSR inverse agonism with PF-5190457 diminished HFD-induced binge-like eating in male and female WT mice, but not GHSR-KO male mice. a** In male mice, repeated exposure to HFD led to an increase in 2-hour HFD-intake (##*p* < 0.01), but icv. PF-5190457 administration diminished this effect and reduced 2-hour HFD intake compared to icv. vehicle group on Days 2-4 (***p* < 0.001; ****p* < 0.0001; vehicle: *n* = 15, PF-5190457: *n* = 10). **b** In female mice, repeated exposure to HFD led to an increase in 2-hour HFD-intake (#*p* < 0.05), but icv. PF-5190457 administration diminished this effect and reduced 2-hour HFD intake compared to icv. vehicle group (**p* < 0.01; vehicle: *n* = 9, PF-5190457: *n* = 10). **c** In GHSR-KO male mice, repeated exposure to HFD failed to increase 2-hour HFD-intake, and also icv. PF-5190457 did not have any effect.

**Anti-Ghrelin vaccine in rats and mice exposed to HFD**
Previous studies showed that peripheral sequestration of ghrelin by a ghrelin-specific vaccine blunted weight gain in non-obese male Swiss Webster mice[35], non-obese male Wistar rats[36], and pigs[37]. This vaccine is a putative target to manipulate GHSR signaling, but the potential sexual dimorphism of this effect has not been investigated before. We examined the effects of a ghrelin-specific vaccine in the development of obesity using a HFD rat model (males) and subsequently, we expanded this experiment to male and female mice as well. We detected only a modest IgG response after repeated vaccination in rats (Supplementary Fig. S15) and mice (Supplementary Fig. S21), compared to similar studies[35,38–40]. Moreover, antibody plasma ghrelin binding affinity was muted in rats (Supplementary Fig. S16) compared with an earlier study[36]. As such, full affinity analysis was not performed. Consistent with the limited immune response observed, our

anti-ghrelin vaccine did not prevent weight gain (Supplementary Figs. S17 and S22), nor did it lower food consumption (Supplementary Fig. S18 and S23), in experiments with male HFD rats or with male and female Chow/HFD mice. For additional details see Supplementary Information and Supplementary Figs. S19, S20.

## Discussion
Our results present a multifaceted profile of how *GHSR* deletion modulates the effects of long-term (12-month) HFD in male and female rats. The main finding is that the deletion of *GHSR* protects against diet-induced weight gain and decreases food intake during HFD in male but not in female rats. The reduced food intake in our model was not related to changes in locomotor activity or anxiety-like behavior evaluated in the open field and novelty-suppressed feeding tests. All HFD groups, regardless of sex, accumulated more gonadal WAT, but in the males, WT rats had higher relative gonadal WAT weight (% of total body weight) than GHSR-KO. Using infrared thermography, we detected a higher body temperature of the interscapular region, where the BAT is mainly located, in GHSR-KO compared with WT rats in males but not females. Another novel finding of our study was that the HFD decreased brain FDG uptake in males of both genotypes, with a greater effect in WT than in GHSR-KO rats. In females, we observed the opposite, HFD attenuated FDG uptake more markedly but also increased in certain brain areas in GHSR-KO rats. Furthermore, FDG uptake was increased in GHSR-KO male rats compared to WT, regardless of diet, an effect stronger in males than in females. Results of the RNA-sequencing in male rats showed upregulation of gene sets responsible for fat oxidation in the BAT of GHSR-KO and increased expression of connective tissue genes in WT, and these differences were more prominent in HFD. Furthermore, in the WAT during HFD, inflammation-related pathways were upregulated in WT compared to GHSR-KO. We also tested a therapeutically promising GHSR inverse agonist for the first time in both male and female mice. PF-5190457 is the only GHSR blocker that advanced to clinical trials and is well-tolerated in humans, and in our experiments, ICV administration attenuated ghrelin-induced food intake interestingly only in male, but not female mice. The effect of GHSR inverse agonism on a more hedonic aspect of food intake, the HFD-induced binge-like eating, was not sex-dependent; ICV PF-5190457 reduced HFD intake in both sexes.

Our experiments demonstrate that diet-induced weight gain is attenuated in GHSR-KO male rats. The HFD in our model contained almost twice as many calories (5.24 *vs.* 2.89 kcal/g) as the regular diet, less carbohydrates (20% *vs.* 58%), and more fat (60% *vs.* 18%). The significantly reduced weight of GHSR-KO male rats in the HFD-fed group might result from decreased food intake compared to the WT male rats on the same diet. The initial characterization of our model[26] used a regular diet for 34 weeks (*vs.* 48 weeks in the current study) in male rats only and observed a small but significant reduction of body weight and food intake in the KO group. Of note, in the present study, we did not observe a decrease in body weight in rats on the regular diet, only on the HFD. In a previous 19-week study[34] in male and female mice, the lack of GHSR reduced HFD intake and protected against HFD-induced weight gain compared to WT regardless of sex, whereas the lack of GHSR reduced weight in female but not in male mice fed a regular diet[34]. These findings indicate the complex interaction of sex and genetic modulation of the GHSR system in controlling food intake and body weight gain. The significantly reduced food intake observed with HFD but not with a regular diet may involve central mechanisms. GHSR is densely expressed in the rat arcuate nucleus (Arc) and ventromedial hypothalamus (VMH)[10], and HFD can increase GHSR expression in these brain regions[41]. Although earlier studies described reduced locomotor activity after deletion of GHSR[34,42], we did not detect any difference between KO and WT rats in the open field test. Additional locomotion tests, including measuring locomotion in the home cages, may yield different results. The reduced weight gain in GHSR-KO animals might also result from a reduced feed efficiency, like Zigman et al. described earlier in mice[34]. Interestingly, we did not observe similar findings in Wistar rats; however, our animals were older (2-4 months *vs.* 4 weeks) at the beginning of the experiments, which also

lasted significantly longer (12 months vs. 19 weeks), and the genotype-related difference in feed efficiency was less apparent toward the end of the 19-week study[34]. By contrast, we identified the decreased food intake as the main reason for the reduced weight gain in GHSR-KO male rats.

In the current study, GHSR deletion reduced normalized gonadal, but not inguinal, WAT weight (% of total body weight) regardless of diet. This finding may be explained by the lower expression of GHSR in the inguinal WAT than in the gonadal WAT[43]. In rats, inguinal and gonadal fat pads are the main subcutaneous and visceral WAT depots, respectively[44], suggesting that in the present study, GHSR deletion reduced the ratio of visceral fat in the total body weight. The Subcutaneous fat has a more beneficial metabolic profile compared to visceral fat in rodents and humans, by improving insulin action, and longevity, protecting against ectopic fat deposition, lipotoxicity, and reducing tumorigenesis. The higher ratio of subcutaneous fat is also associated with lower triglyceride and higher high-density lipo-protein levels[44]. Deletion of GHSR may exert beneficial effects by reducing visceral (gonadal) WAT weight.

GHSR activity modulates the microstructure, gene-expression profile, and norepinephrine sensitivity of BAT, thereby affecting thermogenesis and energy expenditure. The number of mitochondria and the percentage of multilobular (brown) adipocytes is higher in GHSR-KO male mice[45]. GHSR expression in BAT increases with aging, leading to a thermogenesis reduction[43]. GHSR ablation attenuated the decline of thermogenesis, uncoupling protein-1 expression, and BAT's norepinephrine content in male mice[43]. Consistent with this literature, we found that thermogenesis, measured by intrascapular temperature, was increased in GHSR-KO male rats, regardless of diet.

To gain a deeper understanding of the role of GHSR in BAT thermogenesis, we used RNA-seq to profile gene expression in the BAT in male rats. We found that GHSR gene deletion produced a BAT gene expression program characterized by the upregulation of skeletal muscle-related genes and other oxidative metabolism genes, such as tricarboxylic acid cycle and electron transfer genes, glycolytic genes, fat metabolism and β-oxidation genes, and mitochondrial genes. This effect of GHSR deletion became more pronounced during exposure to the HFD. Such a BAT gene expression profile has elements of the myogenic transcriptional signature that are dynamically regulated[46–48]. BAT and skeletal muscle, both of which contribute to the regulation of body temperature, have common cell lineages[46,47]. The skeletal muscle-like gene expression program in BAT and beige adipocytes is associated with heat generation[49]. Evidence shows that muscle-like actomyosin tensional response in BAT leads to calorie consumption that result in heat production and is induced by β-adrenergic activation[49]. Both BAT and skeletal muscle have high oxidative capacities and utilize glucose and fatty acids as substrates[50,51]. Glycolytic genes, fat metabolism, and β-oxidation genes were also increased in male rats by GHSR deletion. Increased glycolytic activity in beige fat in a myogenic state has also been associated with increased thermogenic capacity[52].

WT male rats showed greater expression of extracellular matrix, integrins, and collagen genes. A late event in BAT gene expression response to HFD is the expression of connective tissue genes, including procollagen, procollagen cleavage, and remodeling genes, which takes place after the upregulation of the skeletal muscle-related gene expression program has subsided[48]. Thus, GHSR appears to direct the BAT gene expression program in male rats, both on the regular chow and on the HFD, possibly acting as a *switch* between a gene expression signature characterized by greater expression of the muscle-related gene expression program and one characterized by the expression of extracellular matrix, integrins, and collagen genes. The present results support GHSR as a therapeutic target that will implement the myogenic gene expression program in BAT, which has been proposed as a therapeutic strategy for obesity and obesity-related metabolic disorders like type 2 diabetes[53]. Enhancing the myogenic gene expression program by blocking GHSR activity seems to increase BAT glucose uptake and insulin sensitivity, which may prevent type 2 diabetes[53]. However, BAT may have a limited role in energy expenditure in humans, especially in obese individuals, where the ratio of BAT activated by cold

exposure is significantly lower[54]. Furthermore, increasing energy expenditure may also activate compensatory mechanisms, that facilitate adaptation to maintain body weight and energy balance[55].

Gene expression analysis of the WAT of GHSR-KO male rats on the HFD *vs.* WT male rats on the same diet showed decreased expression of inflammation-related pathways in GHSR-KO male rats. As diet-induced obesity advances, angiogenesis can no longer keep up with the enlargement of adipose tissue. Pathological processes, such as hypoxia, may cause the dysregulation of adipokines, and adipocyte necrosis, leading to inflammation, fibrosis, and insulin resistance[56]. Furthermore, aging is also associated with a low-grade adipose tissue inflammation[57], and our rats were 14-16 months old so GHSR deletion might be protective against aging-related inflammation[57]. Confirming our findings, macrophage infiltration, as a marker of increased inflammation, was reduced in gonadal WAT of adipose tissue-specific GHSR-KO male mice[58], GHSR deletion promoted a shift of macrophages toward an anti-inflammatory phenotype, and reduces pro-inflammatory cytokine expression in the WAT[57]. Adipose tissue inflammation is reduced by interventions that improve fat oxidation and obesity[59–61].

The differences in BAT/WAT gene expression and BAT thermogenesis described above are important components of GHSR deletion's protective effect in HFD. Not only is food intake reduced by GHSR deletion, but also lipid metabolism is increased, marked by a lower respiratory quotient[34]. In our study, we did not measure respiratory quotients, but it would be interesting to examine it in our genetic rat model or after ICV administration of GHSR blockers. The increased metabolism and energy expenditure were also observed when GHSR was deleted only in neurons[41], suggesting that GHSR activity affects adipocyte metabolism via the CNS. ICV ghrelin administration decreases, whereas neuronal GHSR deletion enhances thermogenesis in brown adipocytes[41,62]. In addition to hypothalamic regions, such as Arc, paraventricular nucleus (PVN), and ventromedial hypothalamus, the VTA also plays a role in GHSR-mediated central regulation of thermogenesis[41]. In white adipocytes, ICV ghrelin infusion decreased the expression of lipid oxidation enzymes, whereas it increased the expression of fat storage-promoting enzymes[62]. The above-mentioned reduction of gonadal WAT in GHSR-KO rats might be regulated also centrally, as it was shown previously that ICV ghrelin treatment increased the weight of gonadal WAT[63]. To further describe the role of constitutive GHSR activity in central regulation of adipocyte metabolism, experiments with ICV administration of the inverse agonist, PF-5190457 are required.

An additional finding of our study is the lipidomics analysis, based on an unbiased approach, to investigate the effect of HFD and genotype on the lipid profile of WAT and BAT. It is known that dietary fatty acid intake can cause changes in lipid composition. The HFD-fed rats in this study were high in saturated fatty acids, especially for triglyceride species[64]. We detected increased saturated and decreased unsaturated triglycerides in BAT and WAT due to HFD in both sexes. A recent study[65] using an untargeted lipidomic approach for HFD-induced obesity in male rats showed a similar pattern in plasma, gonadal WAT, and liver. In our study, GHSR deletion affected the TG profile only in the WAT in males, leading to increased triglycerides with unsaturated fatty acids. This genotype-related difference was observed only with a regular diet, which can be explained by the robust decreasing effect of HFD on unsaturated triglycerides. We speculate that this effect of diet may have overshadowed the genotype effect.

Ghrelin can stimulate hepatic lipogenesis[66,67], whereas inverse agonism of GHSR is preventive against hepatic steatosis in DIO mice[68], and GHSR antagonism enhances hepatic fatty acid oxidation in pigs[69]. We investigated the hepatic steatosis in our Wistar rats using histology and quantitatively by measuring hepatic triglycerides. We could not detect any effect of GHSR deletion on hepatic steatosis in both sexes. However, we established that HFD increased the concentration of hepatic triglycerides more in males than females. This is consistent with the epidemiology of non-alcohol-related fatty liver disease (NAFLD) and non-alcohol-related steatohepatitis (NASH), which show higher prevalence in males[70]. Although future studies

will reveal the interactions between sex, genetics, and environmental factors behind hepatic steatosis, several pathways have been identified as potentially playing a role in sex differences. For example, estrogens may affect fatty acid transport, esterification, de novo lipogenesis, and immune responsiveness in the liver[71–73].

The small animal PET highlighted differences in brain FDG uptake 50 weeks after initiation of a regular or high-fat diet in WT and GHSR-KO rats. HFD decreased brain FDG uptake, an observation consistent with early work in male Wistar rats[74], male Sprague-Dawley rats[75], and male C57BL/6J mice[76]. Yet, this work is novel not only because we investigated sex differences but also for the length of HFD exposure, as previous studies measured FDG uptake only after 9[75,76], or 16 weeks[74], and also for the brain-region specific analysis, described in Supplementary Information. The brain glucose uptake by glucose transporters is also affected by HFD, as GLUT-4 protein levels were decreased after 9 weeks of HFD in male C57BL/6 J mice[76]. GHSR gene deletion diminished the negative effect of HFD on brain FDG uptake in male rats and increased FDG uptake regardless of diet. GHSR may affect the glucose transporters on neurons (GLUT-3) and on astrocytes (GLUT-2)[77], as Fuente-Martin et al. showed that both acute and chronic central ghrelin administration decreased hypothalamic GLUT-2 and GLUT-3 protein levels in vivo in male Wistar rats, and GHSR gene deletion abolished this effect in vitro[77]. In female rats, we observed the opposite, namely HFD markedly decreased brain FDG uptake in GHSR-KO, but not in WT rats. This result is consistent with a recent study that showed no changes in brain FDG uptake after 16 weeks of HFD in females, but a decrease in male Wistar rats[74]. In addition, the main effect of Genotype (GHSR-KO > WT) on brain FDG uptake was less prominent in females.

In the FDG uptake experiment, we identified brain areas where glucose metabolism was affected by the deletion of *GHSR*. It is known that areas related to reward, gustatory, olfactory, or visuospatial sensation processing, show altered basal activity in obesity[78–83]. In our study, in male rats, *GHSR* deletion diminished the FDG uptake decreasing the effect of HFD in important brain areas such as the nucleus accumbens (ventral striatum), dorsal striatum, and basal amygdaloid nucleus. In females, FDG uptake in the dorsal striatum was increased in HFD *vs.* chow in KO females, whereas remained unaffected by diet in WT females, suggesting a genotype difference. In males, WT rats showed decreased FDG uptake in the motor and orbital cortices regardless of diet, whereas in females the same was observed only during HFD in the motor cortex. In the somatosensory cortex, HFD was associated with a decreased FDG uptake, but this effect was attenuated in KO rats. Our findings are consistent with human structural brain changes reported in obesity, namely reduction in grey matter in inferior frontal and precentral gyri, amygdala, dorsal striatum (caudate nucleus and putamen), and ventral striatum[84]. The piriform cortex is the largest component of the primary olfactory cortex, playing a key role in processing olfactory cues[85]. In the piriform cortex, in male rats, we observed higher FDG uptake in KO animals regardless of diet and in the chow diet compared to HFD regardless of genotype. In female rats, HFD was associated with increased FDG uptake in WT animals and with decreased uptake in KO animals.

Sensory, mnemonic/cognitive, and interoceptive (endocrine and vagal signals) control of food intake and energy balance all converge to the hippocampus[80], where GHSR is densely expressed[86]. In the entorhinal cortex, FDG uptake was increased in GHSR-KO male rats compared to WT, regardless of diet; and was also increased by HFD in KO rats and remained unaffected by diet in WT. In the hippocampus CA fields, KO rats showed increased FDG uptake during a regular diet, but HFD diminished this effect by decreasing FDG uptake regardless of genotype. KO rats during regular diet showed increased FDG uptake in the dentate gyrus as well, but HFD diminished this effect by increasing FDG uptake in WT rats. Genotype differences were observed in females too, as FDG uptake was increased by HFD in the entorhinal cortex only in WT female rats; HFD increased FDG uptake in the hippocampus CA fields in WT rats, whereas it remained unaffected in KO; and KO females showed decreased FDG uptake during HFD. The different patterns of FDG uptake of these three hippocampal regions (entorhinal cortex, hippocampus CA fields, and dentate gyrus) and their sex differences could be targets of further research.

This Wistar GHSR-KO model was previously characterized in males[26]. In mice, GHSR deletion was protective against HFD-induced weight gain in both sexes[34]. Sexual dimorphism of the ghrelin system has been described; for example, Sprague-Dawley female rats[87] had higher plasma ghrelin levels, lower LEAP2 expression in the liver, and higher GHSR expression in the hippocampus, amygdala, and LH. Gonadal hormones may be responsible for this sexual dimorphism given that ovariectomy diminished, and estrogen replacement restored the differences in ghrelin levels between males and females[87]. In our Wistar rats, we did not find any difference in ghrelin or desacyl-ghrelin levels between females and males, regardless of genotype, but LEAP2 plasma concentration was higher in males. Furthermore, estrogen decreases the sensitivity to ghrelin's orexigenic effects, as males and ovariectomized females were more responsive to peripheral and central ghrelin administration than intact or estrogen-treated ovariectomized Long-Evans rats[88].

The terminal hormone analysis established that chow-fed GHSR-KO male rats had higher peripheral levels of progesterone compared to chow-fed WT males, and HFD-fed GHSR-KO males. To date, the effects of ghrelin and GHSR on progesterone levels have only investigated in female rats[89]. Ghrelin administration reduced progesterone levels throughout the estrous cycle in rats[90]. The effect of obesity was examined in human males, in that there was a negative correlation between plasma progesterone levels and body weight, body mass index, and waist circumference[91]. Interestingly, acyl-ghrelin and LEAP2 levels were not different due to Diet or Genotype. Numerous studies reported higher LEAP2 and lower acyl-ghrelin levels after HFD exposure[92–94], but our exposure was quite prolonged compared to those studies (12 months vs. 12-16 weeks), which may have led to adaptations in ghrelin and LEAP2 secretion. In an earlier study, after 60 weeks of HFD in rats, basal ghrelin levels were slightly decreased, but the fasting-induced ghrelin levels were not different[95].

We amended our comprehensive characterization of GHSR-KO Wistar rats with central pharmacological interventions targeting GHSR in male and female C57BL/6 J mice. PF-5190457 is a promising compound, the first GHSR blocker ever tested in clinical trials. This inverse agonist was found to be safe and tolerable and reduced food-seeking in humans[32,38]. Our novel finding was that its effect on ghrelin-induced food intake is sex-dependent, and can be observed only in males, similar to what we described here in GHSR-KO rats and DIO. In the arcuate nucleus of the hypothalamus (Arc), NPY/AgRP neurons are responsible for ghrelin-induced food intake[96]. Rats lacking GHSR only in Arc neurons have lower body weight, less adipose tissue, and reduced daily food intake compared to WT, and these effects could not be restored by ghrelin treatment[97]. PF-5190457 also reduced this activation as measured by the number of cFos-IR cells in the Arc. We tested PF-5190457 in a HFD-induced binge-like eating paradigm, which models how palatable foods induce hedonic food intake and hyperphagia[28–30]. ICV PF-5190457 was effective in both sexes, and we could confirm that this effect is GHSR-dependent. This finding highlights the role of GHSR constitutive activity in binge eating, which involves the GHSRs in reward-related regions like the VTA. Intra-VTA ghrelin injection induces dopamine release in the nucleus accumbens[98,99], and enhances the consumption of palatable foods, whereas intra-VTA injection of a GHSR antagonist reduces HFD intake, but not the intake of a less palatable food[100,101]. Dopaminergic neurons in the VTA were later identified as being responsible for GHSR effects on binge eating and hedonic food intake[30]. GHSR activity can also stimulate the mesocorticolimbic dopamine pathway indirectly via other GHSR-expressing brain areas such as the laterodorsal tegmental area or the lateral hypothalamus[10,102]. Our route of administration (ICV) targeted only GHSRs expressed in the CNS, therefore conducting experiments with systemic administration (e.g., intraperitoneal) of PF-5190457 are required in the future not only to assess the role of peripheral GHSRs in the mechanisms mentioned above but also because of its translational value. Furthermore, examining the effects of PF-5190457 on ghrelin-induced food intake and HFD-induced binge-like eating in rats

would also expand our knowledge about this important GHSR inverse agonist.

Study limitations include that our endocrine outcome was a terminal hormone analysis at a single time point, before euthanasia. Results of the endocrine assays could be influenced by the actual phase of the estrous cycle in females, which was neither controlled nor identified at the time of the sample collection. All of our rats were 14-16 months old at the time of endocrine assays, or sample collection for RNA sequencing, which limits our ability to gain information about the effects of GHSR deletion in a younger age, and we also do not have information about the timeline of how hormonal levels, gene expression patterns changed during the whole experiment. Having a single timepoint for measuring concentrations of all investigated hormones does not take the circadian rhythm into account, which is also a limitation in interpreting our results (especially for corticosterone, GH, aldosterone, testosterone, and progesterone). For example, we detected very low levels of GH, thus the lack of effect of GHSR deletion on GH levels could be due to its pulsatile secretion pattern, and our timepoint was not at the peak (e.g., a floor effect). A further limitation is that we could not measure the binding affinity of the antibody in mice treated with the anti-ghrelin vaccine. Of note, we employed, herein, a global KO model and this approach holds limitations related to long-term adaptations that may occur in constitutive KO rodents[103]. Nonetheless, this approach holds important value, especially as a proof-of-concept related to the role of GHSR in a translationally valid model of obesity and its relevance for genetic mutations in ghrelin systems in humans, such as single nucleotide polymorphism[104]. Of note, albeit in a different context and phenotype, i.e., people with alcohol use disorder (for a review, see Farokhnia et al.[105]), recent human work suggests that GHSR inverse agonism leads to a reduction in food-seeking behaviors[32].

In conclusion, we demonstrated a protective effect of GHSR deletion in diet-induced obesity in male, but not female, rats, which shows a dramatic sexual dimorphism. Our pharmacological studies in mice also revealed that the GHSR blockade has stronger effects in males than in females indicating that the pharmacological intervention of the GHSR in humans requires a precise assessment of the sex effects. Our results indicate an important role for GHSRs in the regulation of body weight, food intake, energy homeostasis, adipose tissue, and brain activity, and constitute a promising target for medication development for the treatment of obesity.

## Methods
### Animals
WT and GHSR-KO rats were obtained from the National Institute on Drug Abuse Intramural Research Program (NIDA IRP) breeding facility. Originating from a Wistar background, the initial characterization of these rats has been previously described[26]. A total of 61 rats were used. We used 28 male (13 WT; 15 GHSR-KO) and 33 female (16 WT; 17 GHSR-KO) rats that were 2-4 months old and weighed 150-350 g at the beginning of the study. Their body weight was recorded twice weekly for 12 months. All animals were single-housed and maintained under a 12 h/12 h light/dark cycle (lights off at 18:00) at 21 °C ± 2 °C. All procedures were conducted in accordance with the National Institutes of Health Guide for the Care and Use of Laboratory Animals and were approved by the NIDA IRP Animal Care and Use Committee, and IACUC of Multidisciplinary Institute of Cell Biology (IMBICE, National University of La Plata, Buenos Aires, Argentina). We have complied with all relevant ethical regulations for animal use.

### Diets
Rats were fed either a regular chow diet (Chow) with 24% protein, 58% carbohydrate, 18% fat (2.89 kcal/g; 2018 Teklad ENVIGO) or a HFD with 20% protein, 20% carbohydrate, 60% fat (5.24 kcal/g; Research Diets D12492, New Brunswick, NJ) for a total of 12 months (52 weeks). Animals were randomly assigned either Chow or HFD at the beginning of the study. For the duration of the study, rats had *ad libitum* access to food and water, except during periods of fasting required for the study procedures and behavioral testing described below.

### Body weight and food consumption
Body weight (BW, expressed in g) and food intake (expressed in g per week) were recorded twice a week. Behavioral assessments (Supplementary Information) and endocrine assays (Supplementary Information) were also performed.

### Carcass analysis
The rats had *ad libitum* access to food and water prior to euthanasia. Experimenters were blind to both the genotype of the rats and the diet that they received. The rats were weighed, and body measurements and temperature were taken. These measurements were followed by decapitation under deep isoflurane anesthesia. Brain, liver, spleen, and adrenal glands were collected and weighed. White adipose tissue (WAT), including visceral fat pads (gonadal, inguinal), and intrascapular BAT were dissected and weighed. The carcass analysis was adapted from a previously published protocol[106] and was consistent with the carcass analysis described in the initial development and characterization of our GHSR-KO rat model[26].

### Brown Adipose Tissue (BAT) Thermogenesis
At the end of the study (week 52), surface body temperature was determined using an infrared thermometer (FLIR E60). Rats were anesthetized, shaved at the rear base of the ears and center on the scapulae and positioned below the infrared thermometer. The thermometer was held at a focal length of 30 cm above the interscapular region where BAT is located underneath the skin of the rats.

### Small animal PET
Previous studies have used positron emission tomography (PET) with [18F] fluorodeoxyglucose (FDG) to measure the brain's metabolic response to an array of food stimuli and obesity[107]. Here, we used small animal PET to measure brain glucose uptake 50 weeks after initiation of Chow or HFD. Six rats per group were selected at random and fasted overnight for 12 h prior to FDG administration.

The PET scans were adapted from a previously published protocol[108]. Briefly, the food was removed 1 h before the beginning of the dark cycle; the rats had *ad libitum* access to water. Acute food deprivation is necessary for PET with FDG because circulating glucose competes with FDG uptake. On the day of the scan, rats were injected with FDG (1 µCi/g/BW, i.p.) and allowed 30 min for uptake in their home-cages. The rats were then placed in a sealed anesthesia induction chamber and anesthetized with 4-5% isoflurane mixed with oxygen. Next, the anesthetized rats were placed in a prone position on the scanner bed. Glucose was measured via a tail blood sample using a commercially available Aimstrip Plus blood glucose monitor (Germaine Laboratories, San Antonio, TX). Anesthesia was maintained with 2% isoflurane and rats were scanned for 20 min using a Mediso nanoScan PET/CT scanner. During the scan, the rats were monitored visually by the experimenter, and scanner beds were equipped with respiratory and body temperature sensors. After the scan, rats were kept in a holding area for 24 h for FDG to decay. Rats were returned to their regular holding rooms in clean cages after radiotracer activity had completely decayed (measured via survey meter and activity counter).

### Molecular RNA-Seq analysis of the adipose tissue
Given our previous work in non-obese GHSR-KO rats[26] and the results of the BAT thermogenesis and body temperature (present study; see **Results**), adipose tissue from the carcass analysis was used for an unbiased hypothesis-generating analysis using gene expression and pathway analysis by RNA-seq and lipidomic analysis by mass spectrometry.

Gene expression was profiled by RNA-Seq. We used Gene Set Enrichment Analysis (GSEA) for pathway analysis[33] in conjunction with genesets from the Molecular Signatures Database (MSigDB). Libraries were generated for next-gen sequencing following the protocol recommended in the ScriptSeq v2 RNA-Seq Library Preparation Kit (Illumina), and uniquely barcoded with the ScriptSeq Index PCR Primers (Illumina) to generate ~30 M 1×75 bp reads/sample. RNA-Seq reads were aligned and quantified

to the RefSeq rat transcriptome (dated 2021-05-26) using RSEM (version 1.30)[109] with Bowtie2[110]. Differential expression and log2 fold changes were calculated with DeSeq2[111]. Gene set enrichment calculations were completed using GSEA (v. 4.0.3)[33] with statistics estimated by 10 K gene set permutations and the log2 fold changes of the 8000 genes with the lowest unadjusted p-values from DeSeq2. The Broad-UC San Diego Molecular Signatures Database (MsigDB v7.4) curated canonical pathway set was used (http://www.gsea-msigdb.org/gsea/index.jsp), along with the v7.4 mapping to human orthologs.

### Lipidomic analysis of the adipose tissue

From the tissue harvested during carcass analysis, brown adipose tissue (BAT) and white adipose tissue (WAT) samples were homogenized using a bead mill homogenizer (Bead Ruptor 12, Omni International, Kennesaw, GA). For homogenization, 500 μL of ice-cold water: and ethanol was added per 10 mg of tissue. Two hundred and fifty μL of homogenate (5 mg of tissue) was placed in a glass extraction vial. Next, 1250 μL of chloroform/methanol (2:1, v/v) was added to the sample followed by 10 μL of the internal standard (triacylglycerol (TG) 30:0 at 8.5 mg/mL in chloroform/methanol [2:1, v/v]). Samples were sonicated for 10 min, and then vortexed for 1 h. Next, 250 μL of water was added to the sample. The resulting mixture was vortexed for 1 min and centrifuged at 15200 rpm at 4 °C for 10 min. The lower phase of the mixture (lipid/triacylglycerol phase) was collected and evaporated to dryness using nitrogen. The samples were resuspended in 2.5 mL of chloroform/methanol (2:1, v/v) and stored at −80 °C until mass analysis.

For mass spectrometry, 10 μL of the lipid extract was diluted in 990 μL of 10 mM of ammonium acetate in methanol. Samples were analyzed on an Orbitrap Velos (Thermo Fisher, San Jose, CA) coupled with an auto sampler (Ultimate 3000 HPLC, Thermo Fisher). Ten μL of the samples were injected in the instrument and analyzed in positive ion mode with the heated electrospray ionization (HESI) ion source. The instrument was set in Fourier transform mass spectrometry (FTMS) mode with a mass resolution of 100 K.

### PF-5190457 Administration in C57BL/6J Mice and GHSR-KO Mice

Experiments were performed in male (n = 25) and female (n = 19) C57BL/6J mice and GHSR-KO male mice[34] (n = 11) generated at the animal facility of the Multidisciplinary Institute of Cell Biology (IMBICE, National University of La Plata, Buenos Aires, Argentina). Mice were aged 12-20 weeks, weighed 20-27 g, and fed with regular chow provided by Gepsa (Grupo Pilar, Argentina), which provided 2.5 kcal/g energy (weight composition: carbohydrates 28.8%, proteins 25.5%, fat 3.6%, fibers 27.4%, minerals 8.1%, and water 6.7%). All procedures were approved by the IACUC of the IMBICE (Approval number 10-01-22).

Ghrelin (Global Peptides, cat. PI-G-03) was dissolved in sterile artificial cerebrospinal fluid (aCSF, 300 pmol/uL) and prepared fresh each experimental day. PF-5190457 (Sigma, cat. PZ0270) was dissolved in dimethyl sulfoxide (DMSO) and diluted in aCSF to obtain a final concentration of DMSO lower than 5% v/v to avoid toxicity. One week before the experiment, anesthetized mice were stereotaxically implanted with a single indwelling guide cannula (P1 Technologies) into the lateral ventricle (placement coordinates: AP: −0.34 mm, ML: +1.00 mm and DV: −2.30 mm) and single-housed. Three days before the experiment, mice were daily ICV-injected with aCSF to acclimate them to handling.

During the ghrelin-induced food intake experiments, animals were housed with a limited amount of bedding to visualize chunks of pellets easily. On the experimental day, all food pellets were removed from the food hoppers, and bedding was confirmed to be free of chow remains. Then, mice were first icv-injected with 2 μL of either vehicle (DMSO 5% v/v in aCSF) or PF-5190457 (100 pmol/mouse). 15 min after pre-treatment, mice were ICV-injected with vehicle (aCSF) or ghrelin (20 pmol/mouse) and exposed to a pre-weighed chow pellet on the floor of the home cages. Chow pellets were weighed at 30-, 60- and 120-min after treatment using a calibrated scale (precision = 1 mg), and food intake was calculated by subtracting the weight of the remaining pellet from the initial weight. After the 120-min of acute food intake assessment, mice were left with a pre-weighed quantity of chow pellets to assess overnight food intake. The next morning (at 9.00 – 10.00 am), the remaining chow pellets were weighed to calculate overnight food consumption. A subset of animals was anesthetized and transcardially perfused with heparinized phosphate-saline buffer (PBS) and formalin (4% v/v in PBS) 120 min after ghrelin/vehicle treatment, and c-Fos immunostaining was conducted. These procedures were carried out as described earlier[112].

To assess the effect of ICV PF-5190457 on HFD-induced binge-like eating in WT and GHSR-KO male mice, we used the following protocol. In the morning (9.00 – 10.00 am) of experimental day 1, chow pellets were removed from food hoppers, and mice were ICV-injected with 2 μL of either vehicle (DSMO 5% v/v in aCSF) or PF-5190457 (100 pmol/mouse). Fifteen-min after ICV administration, mice were exposed to a pre-weighed pellet of HFD (Gepsa; HFD pellets provided 3.9 kcal/g energy and their percentage weight composition was: carbohydrates 22.5%, proteins 22.8%, fat 21.1%, fibers 23.0%, minerals 5.6% and water content 5.0%). Mice were left with access to the HFD pellet for 120-min and then HFD pellets were removed from home cages and chow was re-added to the food hoppers. The same procedure was repeated for 4 consecutive days.

### Statistics and reproducibility

All data are expressed as means and standard error of the mean (SEM). Data were analyzed using analysis of variance (ANOVA) with or without repeated measures (details are provided in the Results section). Mauchly's test was used to assess the assumption of sphericity. If that was violated for the main effect of time, the degree of freedom was corrected using Greenhouse-Geisser estimate of sphericity. Post hoc comparisons were performed when appropriate using the Bonferroni test at repeated measures ANOVA, otherwise using Fisher's LSD test. Statistical significance for all analyses was set at p < 0.05. GraphPad Prism 9.1.0 and SPSS were used for statistical analysis. Body weight, glucose tolerance, and insulin tolerance data from different genotypes of the same sex and on the same diet were also compared with each other to reflect the effect of GHSR deletion on body weight more precisely. 2-way repeated measures ANOVA (Time, Genotype) was used in addition to the initial 3-way repeated measures ANOVA (Time, Diet, Genotype).

For the small animal PET experiments, scans were reconstructed on software provided by Mediso. Scans were then co-registered with PMOD (Zürich, Switzerland) to an MRI rat template to ensure all scans were in the same coordinate space. Voxel-wise comparisons were made with Statistical Parametric Mapping (SPM12, London, UK) to assess the whole brain between group differences in FDG uptake. A two-way (Genotype x Diet) ANOVA was used followed by post hoc comparisons to assess group differences. For each sex, diet groups were combined to compare how GHSR gene deletion altered FDG uptake. A two-sample t-test was used to test both increases and decreases in FDG uptake between WT and GHSR gene deletion. Cluster plots (k = 100) show differences between subjects, with all statistical significance set at p = 0.05.

For the lipid assignment and lipidomic data analysis, a total of 43 triacylglycerols (TGs) were detected as $[M + NH_4]^+$ mass peaks and assigned with a mass error <6 ppm. Mass peaks for TGs were labeled as follows: species number equals the total carbon length and number of carbon double bonds of the acyl chains. R software (version 2.15.1) was used to annotate the data and report the peak intensities of TG species that were normalized with the peak intensity of the internal standard. All data were expressed as means and standard errors of the mean (SEM). A two-way (Genotype x Diet) ANOVA was used followed by Tukey's multiple comparison test to assess group differences. Statistical significance for all analyses was set at p < 0.05. GraphPad Prism 8.0.1 was used for statistical analysis.

The Methods section and the Supplementary Materials and Methods include adequate experimental and characterization data to enhance the reproducibility of our experiments.

## Reporting summary

Further information on research design is available in the Nature Portfolio Reporting Summary linked to this article.

## Data availability

The source data behind the main graphs in the paper (Figs. 1, 4, 7, 8) is available in Supplementary Data 1. RNA sequences are available in the European Nucleotide Archive (ENA), with the accession code PRJEB74923. Other data supporting the findings of this study are available from the corresponding author upon request.

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

## Acknowledgements

We would like to thank Dr. Brendan Tunstall, Dr. Lia Zallar, Caroline Smith, and Dr. Francois Vautier, NIDA IRP, for technical and administrative support. We also thank the team of the Yale Center for Genome Analysis (Yale University, New Haven, CT) for technical and administrative support and Dr. Gail Seabold (NIDA) for proofreading the manuscript. This work was supported by the NIDA IRP (AHL, AGF, RCNM, JLG, JCMV, VC, SLD, SNJ, ZBY, GFK, MM, LFV, LL) and the NIAAA DICBR (AHL, AGF, VC, SLD, LFV, LL). Additional support was also provided via funding HHSN2757015000061 (PI: KDJ). AHL and RCNM were fellows from the Center on Compulsive Behaviors, NIH (2018-2023). MPC and MP was supported by Fondo para la Investigación Científica y Tecnológica (FONCyT, PICT2017-3196, PICT2019-3054 and PICT2020-3270 to MP).

## Author contributions

L.F.V. and L.L. initiated, designed, and supervised the study and shared senior authorship. A.G.F., R.C.N.M., J.C.M.V, V.C., S.L.D. and K.E.W. conducted most of the experiments and acquired data. A.H.L., A.G.F. and R.C.N.M. analyzed data and wrote the manuscript and supplements. G.F.K., L.F.V. and L.L. revised the manuscript and supplements. A.H.L., A.G.F. and R.C.N.M. share first authorship. A.H.L. is the first co-first author because of analyzing and interpreting data, planning and coordinating the writing and revision process, writing the discussion, and finalizing the manuscript. J.L.G. and M.M. designed, conducted, and analyzed the small animal PET. RNA Sequencing was managed and analyzed by V.R.C., B.Z., P.S. and P.P.S. S.N.J. run the lipidomic analysis. M.P.C. and M.P. designed, conducted, and analyzed the PF-5190457 administration experiments. Z.B.Y. conducted some of the endocrine assays. K.D.J. designed and supervised the anti-ghrelin vaccine preparation and experiments. M.E. analyzed the liver histology data. K.R. conducted the hepatic triglyceride measurement. All authors contributed to the content of the paper and the development of the final draft and approved the final submitted version.

## Funding

## Competing interests

The authors declare no competing interests.

## Additional information

[1]Clinical Psychoneuroendocrinology and Neuropsychopharmacology Section, Translational Addiction Medicine Branch, National Institute on Drug Abuse Intramural Research Program and National Institute on Alcohol Abuse and Alcoholism Division of Intramural Clinical and Biological Research, National Institutes of Health, Baltimore, MD, USA. [2]Center on Compulsive Behaviors, Intramural Research Program, National Institutes of Health, Bethesda, MD, USA. [3]Neurobiology of Addiction Section, National Institute on Drug Abuse Intramural Research Program, National Institutes of Health, Baltimore, MD, USA. [4]Biobehavioral Imaging and Molecular Neuropsychopharmacology Unit, Neuroimaging Research Branch, National Institute on Drug Abuse Intramural Research Program, National Institutes of Health, Baltimore, MD, USA. [5]Department of Immunology and Microbiology, The Scripps Research Institute, La Jolla, CA, USA. [6]Translational Analytical Core, National Institute on Drug Abuse Intramural Research Program, National Institutes of Health, Baltimore, MD, USA. [7]Grupo de Neurofisiología, Instituto Multidisciplinario de Biología Celular (IMBICE), Universidad Nacional La Plata (UNLP), Consejo Nacional de Investigaciones Científicas y Técnicas (CONICET) y Comisión de Investigaciones Científicas de la Provincia de Buenos Aires (CIC-PBA), La Plata, Argentina. [8]Molecular Targets and Medications Discovery Branch, National Institute on Drug Abuse Intramural Research Program, National Institutes of Health, Baltimore, MD, USA. [9]Pathology Service, Division of Veterinary Resources, Office of Research Services, National Cancer Institute, National Institutes of Health, Bethesda, MD, USA. [10]Research Service, Veterans Affairs Nebraska-Western Iowa Health Care System, Omaha, NE, USA. [11]Department of Biochemistry and Molecular Biology, University of Nebraska Medical Center, Omaha, NE, USA. [12]Department of Internal Medicine, University of Nebraska Medical Center, Omaha, NE, USA. [13]Department of Chemistry, The Scripps Research Institute, La Jolla, CA, USA. [14]Department of Pediatrics, Dan L. Duncan Cancer Center, Baylor College of Medicine, Houston, TX, USA. [15]Stress and Addiction Neuroscience Unit, National Institute on Drug Abuse Intramural Research Program and National Institute on Alcohol Abuse and Alcoholism Division of Intramural Clinical and Biological Research, National Institutes of Health, Baltimore, MD, USA. [16]Center for Alcohol and Addiction Studies, Department of Behavioral and Social Sciences, School of Public Health, Brown University, Providence, RI, USA. [17]Division of Addiction Medicine, Department of Medicine, School of Medicine, Johns Hopkins University, Baltimore, MD, USA. [18]Department of Neuroscience, Georgetown University Medical Center, Washington DC, USA. [19]These authors contributed equally: András H. Lékó, Adriana Gregory-Flores, Renata C. N. Marchette. [20]These authors jointly supervised this work: Leandro F. Vendruscolo, Lorenzo Leggio. ✉e-mail: leandro.vendruscolo@nih.gov; lorenzo.leggio@nih.gov

