## [Peer Review File · Communications Biology]

Reviewers' comments:

Reviewer #1 (Remarks to the Author):

Leko, Gregory-Flores, Marchette et al describe the differences between male and female rats/mice under high-fat diet (HFD) conditions, and how the deletion of GHSR or the inverse agonist PF-5190457 reduce the effects of HFD. Using RNA sequencing, lipidomic analysis, and behavioral tests, the authors describe the sex-specific differences in ghrelin receptor signaling. The results presented in the manuscript are relevant not only to researchers in the field of obesity and metabolic disorders but also to a broader audience interested in sex-specific mechanisms of metabolic regulation. The topic of the paper is interesting and potentially suitable for Communications Biology. However, further analyses and discussion is needed.

1. The results from figures 2 and 3 used PET scans to measure glucose uptake. These results are interesting but insufficiently analysed. A more detailed analysis should be done and included in the figures. It would be interesting to know which areas of the brain are involved in the FDG uptake in GHSR-KO and wt rats. Additionally, a discussion of those brain areas would be of interest.

2. Discuss the mechanisms by which knocking out GHSR might elevate metabolism. Is there a correlation with the brain areas shown with the PET scans? Discuss about the brain connection, while using icv injections, with the changes observed in BAT and WAT. Are those effects perhaps causally related?

3. Clarify why intracerebroventricular (i.c.v.) injections were chosen for the inverse agonist and address the possibility and implications of systemic intraperitoneal (i.p.) injections. Do intraperitoneal injections produce a similar effect? Discuss potential areas of the brain affected after intracerebroventricular (i.c.v.) injections. Maybe a PET scan after injections of the inverse agonist would be interesting, would it affect similar brain areas than GHSR-KO mice?.

4. GHSR deletion induced a specific oxidative and myogenic gene signature in BAT of male rats. Discuss potential challenges in therapies targeting GHSR considering the differences in gene expression observed in BAT and WAT.

5. In Figure 8, PF-5190457 diminished HFD-induced binge-like eating in male and female WT mice, but not GHSR-KO male mice. What about GHSR-KO female mice? How does this compare with the sex differences seen in the previous experiments involving GHSR-KO rats? Why was PF-5190457 not used in rats?

6. Different aspects like hormonal fluctuations, time of the day for the experiments, reproductive status, genetic background and age-related factors can affect the conclusions of the article. Please discuss.

Minor comment

1. Figure 5: the figure legend says "GSHR" instead of "GHSR" in a few places.

Reviewer #2 (Remarks to the Author):

In the present study, Leko and colleagues evaluated the impact of genetic or pharmacological GHSR blockade with PF-5190457 (a GHSR inverse agonist) on diet-induced obesity in male and female rats. The authors found sexually dimorphic effects with males showing lower food intake after genetic or pharmacological blockade, but not females. In addition, GHSR gene deletion increased thermogenesis

and brain glucose uptake in male rats and modified the effects of high-fat diet (HFD) on brain glucose metabolism in a sex-specific manner. Interestingly, GHSR inverse agonist reduced HFD-induced binge-like eating in both sexes. This is an interesting, well-designed and timely study showing the potential of GHSR as a target against DIO and binge eating. In addition, the present study sheds further light into the sexual dimorphism of orexigenic and adipogenic actions of ghrelin through its receptor, GHS-R. Nonetheless, some specific points require to be amended.

Specific comments

1. Introduction, lines 98-102: it is worth mentioning that GHS-R is expressed in key organs involved in the onset of obesity, such as the hypothalamus (Howard AD et al, *Science* 1998, PMID: 8688086) and the adipose tissue (Rodríguez A et al. *Int J Obes* 2009, PMID: 19238155).

2. Results, lines 147-148: the authors state that GHSR gene deletion protects against DIO, in part, by decreasing food intake during high-fat diet (HFD) in male but not in female rats. To gain further insights into the effect of GHSR gene deletion, it would be interesting to calculate the cumulative feed efficiencies in the experimental animals, since other authors have reported that both female and male GHSR-null mice had lower feed efficiencies (Zigman JM et al. *J Clin Invest* 2005, PMID: 16322794).

3. Results, lines 177-178 and Supplemental Fig. 4: please change the term "lipidosis" by "steatosis". As regards the impact of GHSR signalling on hepatic steatosis, both acylated and desacyl ghrelin stimulate hepatic lipogenesis in male rats (Ezquerro S et al, *Sci Reports* 2016, PMID: 28008992) and deletion of GHSR1a deteriorates the derangement of hepatic clock gene patterns in male mice with DIO (Wang Q et al, *Int J Mol Sci* 2018, PMID: 30322022). It would be very interesting to evaluate the potential interaction of sex and genotype on the expression of factors involved in hepatic lipogenesis.

4. Results, section of "Endocrine assays" and Supplemental Fig. 7: one of the main functions of ghrelin is the control of pulsatile growth hormone (GH) release through the stimulation of its receptor GHSR-1 in pituitary and hypothalamic neurons (Howard AD et al, *Science* 1996, PMID: 8688086). Indeed, decreased GHSR signalling in male *Ghsr*-IRES-Cre mouse model is associated with growth retardation and lack of the usual fasting-induced rise in GH (Peris-Sampedro F et al, *Mol Metb* 2021, PMID: 33798772). The authors did not detect changes in GH levels in male rats (Supplemental Fig. 7F), however it would be interesting to include also the differences between male and female rats as shown for other hormones in Supplemental Fig. 8.

5. Results, section of "Endocrine assays" and Supplemental Fig. 8, and Discussion, lines 409-420: it is surprising that HFD feeding does not affect the levels of ghrelin isoforms or LEAP-2, which are reportedly altered in obesity (Tschöp M et al, *Diabetes* 2001, PMID: 11289032; Pöykkö et al, *Diabetologia* 2005, PMID: 15688209; Lugilde J et al. *Cells* 2022, PMID: 35159134; Ezquerro S et al. *Eur J Endocrinol* 2023; PMID: 37358209).

6. Discussion, lines 293-294: a previous report showed that male and female GHSR-null mice exhibit significant reductions in both respiratory quotient and locomotor activity, especially during the dark cycle (Zigman JM et al. *J Clin Invest* 2005, PMID: 16322794). Please discuss it.

Dear Editor and Reviewers:

Thank you for allowing us to submit a revised version of our manuscript titled 'Genetic or pharmacological GHSR blockade has sexually dimorphic effects in rodents on a high-fat diet' to *Communications Biology*. We appreciate the time and effort you and the Reviewers have dedicated to providing valuable feedback on our manuscript. We are grateful to the Reviewers for their insightful comments on our paper. We have been able to incorporate changes to reflect most of the suggestions provided by the Reviewers.

Below is a point-by-point response to the Reviewers' comments and concerns.

Reviewers' comments:

Reviewer #1 (Remarks to the Author):

Leko, Gregory-Flores, Marchette et al describe the differences between male and female rats/mice under high-fat diet (HFD) conditions, and how the deletion of GHSR or the inverse agonist PF-51090457 reduce the effects of HFD. Using RNA sequencing, lipidomic analysis, and behavioral tests, the authors describe the sex-specific differences in ghrelin receptor signaling. The results presented in the manuscript are relevant not only to researchers in the field of obesity and metabolic disorders but also to a broader audience interested in sex-specific mechanisms of metabolic regulation. The topic of the paper is interesting and potentially suitable for *Communications Biology*. However, further analyses and discussion is needed.

1. The results from figures 2 and 3 used PET scans to measure glucose uptake. These results are interesting but insufficiently analysed. A more detailed analysis should be done and included in the figures. It would be interesting to know which areas of the brain are involved in the FDG uptake in GHSR-KO and wt rats. Additionally, a discussion of those brain areas would be of interest.

We agree. Accordingly, we have amended our Supplement with a more detailed analysis and a new Figure, Figure S7 (See Supplement pages 26-29). The new section in the Supplement describes which brain regions showed different FDG uptake regarding diet or genotype in both sexes. We also discussed these results, highlighting brain circuitries regulating food intake and energy balance (See Manuscript pages 22-23, lines 476-505)

Figure S7. Small animal PET. (A-B) Statistical parametric maps ($p < 0.05$) of [^{18}F] fluorodeoxyglucose (FDG) uptake as a function of Diet ($n = 6/\text{treatment group}$) in WT (A) and GHSR-KO (B) male rats. HFD increased (red) and decreased (blue) FDG uptake in different brain regions. (C-D) Statistical parametric maps ($p < 0.05$) of FDG uptake as a function of Diet ($n = 6/\text{treatment group}$) in WT (C) and GHSR-KO (D) female rats. HFD increased (red) and decreased (blue) FDG uptake in different brain regions. (E-F) Statistical parametric maps ($p < 0.05$) of FDG uptake as a function of Genotype ($n = 6/\text{treatment group}$) in Chow (E) and HFD (F) fed male rats. GHSR deletion increased (red) and

decreased (blue) FDG uptake in different brain regions. **(G-H)** Statistical parametric maps ($p < 0.05$) of FDG uptake as a function of Genotype ($n = 6$ /treatment group) in Chow **(G)** and HFD **(H)** fed female rats. GHSR deletion increased (red) and decreased (blue) FDG uptake in different brain regions.

2. Discuss the mechanisms by which knocking out GHSR might elevate metabolism. Is there a correlation with the brain areas shown with the PET scans? Discuss about the brain connection, while using icv injections, with the changes observed in BAT and WAT. Are those effects perhaps causally related?

Thank you for pointing this out. We agree with this comment. Therefore, we discussed how GHSRs in the CNS mainly regulate the BAT and WAT metabolism and which brain areas expressing GHSR play a role in this process (See Manuscript lines 423-434). We also mentioned the need for further experiments with the inverse agonist PF-5190457 to examine the effects of central constitutive GHSR activity on adipocyte metabolism.

3. Clarify why intracerebroventricular (i.c.v.) injections were chosen for the inverse agonist and address the possibility and implications of systemic intraperitoneal (i.p.) injections. Do intraperitoneal injections produce a similar effect? Discuss potential areas of the brain affected after intracerebroventricular (i.c.v.) injections. Maybe a PET scan after injections of the inverse agonist would be interesting, would it affect similar brain areas than GHSR-KO mice?

We appreciate the opportunity to address your concerns and clarify our experimental design choices regarding using icv injections of PF-5190457 for assessing food intake. We chose icv injections for PF-5190457 because the experiments were focused on investigating food intake, a process predominantly regulated by the brain. Importantly, icv infusion facilitates higher concentrations of the drug in the brain, particularly in regions distantly located from fenestrated capillaries, such as the striatum¹. In addition, icv infusion allows us to use lower doses of PF-5190457, as compared to systemic treatment, minimizing the risk of nonspecific effects (which were confirmed not to exist in our experimental conditions as GHSR-KO mice were unaffected by PF-5190457, in contrast to WT mice). Additionally, the low dose of PF-5190457 required for the icv infusions enables the use of a well-tolerated vehicle (DSMO 5% v/v in aCSF), simplifying the study, as systemic infusion necessitates specific vehicle solutions.

Therefore, given the PF-5190457 experiments included in this paper were meant to be a proof-of-concept study aimed at complementing the work with the GHSR KO experiments, and given that hedonic and ghrelin-induced food intake is centrally driven, we wanted to investigate the role of central GHSR constitutive activity; therefore, we chose the icv route of administration.

According to the Reviewer's comment, we discussed potential brain areas that may be targeted by icv PF-5190457 and playing a role on its effects on ghrelin-induced food intake and HFD-induced hedonic food intake (See Manuscript lines 535-538, 544-550). It is possible that a small animal PET after icv inverse agonist treatment would show similar results to the GHSR deletion, but we also need to consider that 1) the timeframe of the effect is very different (acute vs. continuous); 2) GHSR deletion was global, affecting not only the brain but also the whole body, whereas central inverse agonism is restricted to the CNS. Nonetheless, we certainly agree with the Reviewer that future experiments administering PF-5190457 systematically would be important, especially given their potential translational value, hence we have briefly discussed the need for future experiments of that kind in the Discussion section.

4. GHSR deletion induced a specific oxidative and myogenic gene signature in BAT of male rats. Discuss potential challenges in therapies targeting GHSR considering the differences in gene expression observed in BAT and WAT.

Thank you for pointing this out, we briefly discussed the potential challenges of implementing these findings in human obesity/diabetes therapy (please see Manuscript lines 402-405).

5. In Figure 8, PF-5190457 diminished HFD-induced binge-like eating in male and female WT mice, but not GHSR-KO male mice. What about GHSR-KO female mice? How does this compare with the sex differences seen in the previous experiments involving GHSR-KO rats? Why was PF-5190457 not used in rats?

We appreciate the Reviewer's comments. PF-5190457 diminished HFD-induced binge-like eating in both male and female WT mice. However, the statistical analysis indicated that the main effect of treatment (PF-5190457 vs. vehicle) was more significant in males ($p < 0.001$) than in females ($p < 0.01$). Additionally, the treatment x time interaction was only significant in males ($p < 0.0001$), with males consuming less HFD on days 2, 3, and 4 after PF-5190457 treatment compared to vehicle.

Given that males demonstrated greater sensitivity to the anorectic effects of PF-5190457 in the binge-eating protocol, we tested PF-5190457 in male GHSR-KO mice only. In statistical terms, the robust effect of PF-5190457 in male mice implies that a relatively small sample size is required to reveal significant effects. In females, however, the less pronounced effect of PF-5190457 suggests that a large sample size would be needed to confidently argue that PF-5190457 does not have any effect on HFD binge eating in GHSR-KO female mice. Based on this statistical issue, as well as on the previous pharmacological characterization of PF-5190457 that does not suggest

any non-specific effects of this compound, our Institutional Animal Care and Use Committee did not approve the use of GHSR-KO female mice to confirm findings observed in males.

In regard to the last question, this was due to pragmatic reasons. The PF-5190457 experiments were conducted by co-author Dr. Mario Perello in his lab in Argentina, as an independent project, simultaneously with our GHSR-KO rat experiments. Because we found similar effects, we decided to combine them to provide stronger evidence and create a more compelling article. We agree that future similar experiments should also be done in rats and have briefly mentioned it in the Discussion.

6. Different aspects like hormonal fluctuations, time of the day for the experiments, reproductive status, genetic background and age-related factors can affect the conclusions of the article. Please discuss.

Thank you for pointing this out. Accordingly, we discussed these aspects as limitations of interpreting our results, especially in regard to our endocrine outcomes (See Manuscript lines 557-567)

Minor comment:

1. Figure 5: the figure legend says "GSHR" instead of "GHSR" in a few places.

Thanks for your comment; we corrected this mistake in the text.

Reviewer #2 (Remarks to the Author):

In the present study, Leko and colleagues evaluated the impact of genetic or pharmacological GHSR blockade with PF-5190457 (a GHSR inverse agonist) on diet-induced obesity in male and female rats. The authors found sexually dimorphic effects with males showing lower food intake after genetic or pharmacological blockade, but not females. In addition, GHSR gene deletion increased thermogenesis and brain glucose uptake in male rats and modified the effects of high-fat diet (HFD) on brain glucose metabolism in a sex-specific manner. Interestingly, GHSR inverse agonist reduced HFD-induced binge-like eating in both sexes. This is an interesting, well-designed and timely study showing the potential of GHSR as a target against DIO and binge eating. In addition, the present study sheds further light into the sexual dimorphism of orexigenic and adipogenic actions of ghrelin through its receptor, GHS-R. Nonetheless, some specific points require to be amended.

Specific comments:

1. Introduction, lines 98-102: it is worth mentioning that GHS-R is expressed in key organs involved in the onset of obesity, such as the hypothalamus (Howard AD et al, Science 1998, PMID: 8688086) and the adipose tissue (Rodríguez A et al. Int J Obes 2009, PMID: 19238155).

Thank you for the suggestion; accordingly, we mentioned these important GHSR-expressing organs and brain areas in the Introduction (See Manuscript lines 103-105)

2. Results, lines 147-148: the authors state that GHSR gene deletion protects against DIO, in part, by decreasing food intake during high-fat diet (HFD) in male but not in female rats. To gain further insights into the effect of GHSR gene deletion, it would be interesting to calculate the cumulative feed efficiencies in the experimental animals, since other authors have reported that both female and male GHSR-null mice had lower feed efficiencies (Zigman JM et al. J Clin Invest 2005, PMID: 16322794).

Thank you for pointing this out. We calculated the weekly cumulative feed efficiencies as follows: weight gain [mg] / energy consumed [kcal]. We averaged the weekly feed efficiencies for each month to get a single value per month. Then, we compared the GHSR-KO and WT groups using two-way repeated-measures ANOVAs for animals with HFD and regular diet separately. We did not find a significant main effect of Genotype in our comparisons; the Time effect was significant, and the Time x Genotype interaction was not. Because we observed a significant difference in body weight gain and food intake within the HFD cohort, we included in the manuscript the feed efficiency results from this cohort only (See Manuscript lines 166-168, Supplement page 18, and Supplemental Figure S1). Our results are not consistent with the earlier study of *Zigman et al.*, which found a lower feed efficiency in male and female GHSR-KO mice⁶. We discussed the potential reasons for this incongruency. We pointed out that in our experiments the decreased food intake may be the main reason for the protective effect of *GHSR* deletion against diet-induced weight gain (See Manuscript lines 349-355).

A**B**
Supplemental Figure S1. No differences in weekly feed efficiency (weight gain [mg] / energy consumption [kcal]) between WT and GHSR-KO male and female rats fed with HFD. (A) There was no significant difference between WT and GHSR-KO male rats fed with HFD in weekly feed efficiency. WT: $n = 7$, GHSR-KO: $n = 8$ **(B)** There was no significant difference between WT and GHSR-KO female rats fed with HFD in weekly feed efficiency. WT: $n = 6$, GHSR-KO: $n = 10$.

3. Results, lines 177-178 and Supplemental Fig. 4: please change the term “lipidosis” by “steatosis”. As regards the impact of GHSR signalling on hepatic steatosis, both acylated and desacyl ghrelin stimulate hepatic lipogenesis in male rats (Ezquerro S et al, Sci Reports 2016, PMID: 28008992) and deletion of GHSR1a deteriorates the derangement of hepatic clock gene patterns in male mice with DIO (Wang Q et al, Int J Mol Sci 2018, PMID: 30322022). It would be very interesting to evaluate the potential interaction of sex and genotype on the expression of factors involved in hepatic lipogenesis.

Thanks for your comment. Accordingly, we replaced the term “lipidosis” with “steatosis”. Furthermore, we amended our manuscript with a quantification of hepatic steatosis, namely we measured triglycerides in our liver samples (See Manuscript lines 188-193; Supplement page 25 and Supplemental Figure S6). This was conducted by our collaborator Dr. Karuna Rasineni, whom we added as a co-author to our manuscript. However, ghrelin can stimulate hepatic lipogenesis^{7,8}, inverse agonism of GHSR is protective against hepatic steatosis in DIO mice⁹, and hepatic fatty acid oxidation is induced by GHSR antagonism in pigs¹⁰; we could not detect any effect of *GHSR* deletion on hepatic steatosis indicated by hepatic triglyceride concentration. At the same time, the effect of Sex and Sex x Diet interaction were significant, as HFD-males had increased steatosis compared to HFD-females, regardless of Genotype. This is consistent with the epidemiology of non-alcohol-related fatty liver disease (NAFLD) and non-alcohol-related steatohepatitis (NASH), which shows higher prevalence in males¹¹, and related to the sex difference of fatty acid transport,

esterification, *de novo* lipogenesis, and immune response in the liver¹²⁻¹⁴. We discussed our results about hepatic steatosis in our paper (See Manuscript lines 446-457).

Supplemental Figure S6. Hepatic triglycerides (TG) concentration. The concentration of hepatic triglycerides (mg/g tissue) was higher in HFD compared to chow in both females and males (**** $p < 0.0001$) regardless of Genotype. Males had higher hepatic TG than females (**** $p < 0.0001$), but only with HFD. Female WT-chow: $n = 9$; Female KO-chow: $n = 6$; Female WT-HFD: $n = 5$; Female KO-HFD: $n = 9$; Male WT-chow: $n = 5$; Male KO-chow: $n = 6$; Male WT-HFD: $n = 6$; Male KO-HFD: $n = 7$.

4. Results, section of “Endocrine assays” and Supplemental Fig. 7: one of the main functions of ghrelin is the control of pulsatile growth hormone (GH) release through the stimulation of its receptor GHSR-1 in pituitary and hypothalamic neurons (Howard AD et al, Science 1996, PMID: 8688086). Indeed, decreased GHSR signalling in male *Ghsr*-IRES-Cre mouse model is associated with growth retardation and lack of the usual fasting-induced rise in GH (Peris-Sampedro F et al, Mol Metb 2021, PMID: 33798772). The authors did not detect changes in GH levels in male rats (Supplemental Fig. 7F) however it would be interesting to include also the differences between male and female rats as shown for other hormones in Supplemental Fig. 8. Thank you for the suggestion, we changed the Supplemental Figure S11 (Figure S8 in the earlier version) to include female GH levels. We did not detect any effect of Sex, Diet or Genotype on GH levels. However, there is an important limitation in interpreting our results: we measured shallow levels of GH, possibly because GH has a pulsatile secretion pattern, and our timepoint

was not at the peak, leading to a floor effect when analyzing the results. According to the suggestion from Reviewer #1, we pointed this out among the limitations of our study. We measured nose-anus length in our carcass analysis (see Supplemental Table 1), and we did not find any significant difference between KO and WT rats, suggesting the effect of *GHSR* deletion on GH secretion in our model should be limited.

Supplemental Figure S11. Terminal hormone analysis – ghrelin system and GH in males and females. (A-C) There was no significant difference regarding Sex, Diet, or Genotype in plasma concentrations of ghrelin, desacyl-ghrelin, or GH. (D) Males had higher LEAP2 levels compared to females (**** $p < 0.0001$). Males WT: $n = 6$ Chow, $n = 7$ HFD. Females WT: $n = 7$ (GH: $n = 10$) Chow, $n = 5$ (GH: $n = 6$) HFD. Males GHSR-KO: $n = 7$ Chow, $n = 7$ (GH: $n = 8$) HFD. Females GHSR-KO: $n = 4$ (GH: $n = 7$) Chow, $n = 5$ (GH: $n = 9$) HFD. All data are expressed as mean \pm SEM, circles represent each individual rat.

5. Results, section of “Endocrine assays” and Supplemental Fig. 8, and Discussion, lines 409-420: it is surprising that HFD feeding does not affect the levels of ghrelin isoforms or LEAP-2, which are reportedly altered in obesity (Tschöp M et al, Diabetes 2001, PMID: 11289032; Pöykkö

et al, *Diabetologia* 2005, PMID: 15688209; Lugilde J et al. *Cells* 2022, PMID: 35159134; Ezquerro S et al. *Eur J Endocrinol* 2023; PMID: 37358209).

We agree and were surprised as well. We suspect that some of the surprising results found in our study reflect the fact that we exposed these rats to a quite prolonged (12 months) HFD, hence this quite remarkable chronic high-calorie food exposure may have led to some adaptations over time. Contrary to the studies mentioned above, other studies showed different results. *François et al.* found no difference in ghrelin levels after 13 weeks of HFD in mice, and preproghrelin mRNA expression was even elevated after this HFD exposure¹⁵. In an earlier study of Gomez et al., basal ghrelin levels were slightly decreased after 60 weeks of HFD in male Sprague-Dawley rats, but the fasting-induced ghrelin levels were not different¹⁶. Overall, the majority of the studies investigating LEAP2 and acyl-ghrelin levels in DIO used much shorter HFD exposure (12-16 weeks vs. 12 months in our current study). We have discussed this aspect in the paper (See Manuscript lines 524-529).

6. Discussion, lines 293-294: a previous report showed that male and female GHSR-null mice exhibit significant reductions in both respiratory quotient and locomotor activity, especially during the dark cycle (Zigman JM et al. *J Clin Invest* 2005, PMID: 16322794). Please discuss it.

Thanks for your comment. We included these earlier findings in our discussion (See Manuscript lines 346-349). We did not detect any difference in locomotor activity in the open field test. Still, our methodology was different for measuring locomotion and spontaneous activity and more limited than the other studies with mice and rats^{6,17}.

- 1 Perello, M. & Dickson, S. L. Ghrelin signalling on food reward: a salient link between the gut and the mesolimbic system. *J Neuroendocrinol* **27**, 424-434 (2015). <https://doi.org:10.1111/jne.12236>
- 2 Denney, W. S., Sonnenberg, G. E., Carvajal-Gonzalez, S., Tuthill, T. & Jackson, V. M. Pharmacokinetics and pharmacodynamics of PF-05190457: The first oral ghrelin receptor inverse agonist to be profiled in healthy subjects. *Br J Clin Pharmacol* **83**, 326-338 (2017). <https://doi.org:10.1111/bcp.13127>
- 3 Godlewski, G. et al. Targeting Peripheral CB(1) Receptors Reduces Ethanol Intake via a Gut-Brain Axis. *Cell Metab* **29**, 1320-1333.e1328 (2019). <https://doi.org:10.1016/j.cmet.2019.04.012>
- 4 Lee, M. R. et al. The novel ghrelin receptor inverse agonist PF-5190457 administered with alcohol: preclinical safety experiments and a phase 1b human laboratory study. *Mol Psychiatry* **25**, 461-475 (2020). <https://doi.org:10.1038/s41380-018-0064-y>
- 5 Richardson, R. S. et al. Pharmacological GHSR (ghrelin receptor) blockade reduces alcohol binge-like drinking in male and female mice. *Neuropharmacology* **238**, 109643 (2023). <https://doi.org:10.1016/j.neuropharm.2023.109643>
- 6 Zigman, J. M. et al. Mice lacking ghrelin receptors resist the development of diet-induced obesity. *J Clin Invest* **115**, 3564-3572 (2005). <https://doi.org:10.1172/jci26002>

- 7 Ezquerro, S. *et al.* Acylated and desacyl ghrelin are associated with hepatic lipogenesis, β -oxidation and autophagy: role in NAFLD amelioration after sleeve gastrectomy in obese rats. *Sci Rep* **6**, 39942 (2016). <https://doi.org:10.1038/srep39942>
- 8 Li, Z. *et al.* Ghrelin promotes hepatic lipogenesis by activation of mTOR-PPAR γ signaling pathway. *Proc Natl Acad Sci U S A* **111**, 13163-13168 (2014). <https://doi.org:10.1073/pnas.1411571111>
- 9 Abegg, K. *et al.* Ghrelin receptor inverse agonists as a novel therapeutic approach against obesity-related metabolic disease. *Diabetes Obes Metab* **19**, 1740-1750 (2017). <https://doi.org:10.1111/dom.13020>
- 10 Zhang, H., Yan, X., Lin, A., Xia, P. & Su, Y. Inhibition of ghrelin activity by the receptor antagonist [D-Lys3]-GHRP-6 enhances hepatic fatty acid oxidation and gluconeogenesis in a growing pig model. *Peptides* **166**, 171041 (2023). <https://doi.org:10.1016/j.peptides.2023.171041>
- 11 Vernon, G., Baranova, A. & Younossi, Z. M. Systematic review: the epidemiology and natural history of non-alcoholic fatty liver disease and non-alcoholic steatohepatitis in adults. *Aliment Pharmacol Ther* **34**, 274-285 (2011). <https://doi.org:10.1111/j.1365-2036.2011.04724.x>
- 12 Català-Senent, J. F. *et al.* Hepatic steatosis and steatohepatitis: a functional meta-analysis of sex-based differences in transcriptomic studies. *Biol Sex Differ* **12**, 29 (2021). <https://doi.org:10.1186/s13293-021-00368-1>
- 13 Dong, Q. *et al.* Sex-specific differences in hepatic steatosis in obese spontaneously hypertensive (SHROB) rats. *Biol Sex Differ* **9**, 40 (2018). <https://doi.org:10.1186/s13293-018-0202-x>
- 14 Palmisano, B. T., Zhu, L. & Stafford, J. M. Role of Estrogens in the Regulation of Liver Lipid Metabolism. *Adv Exp Med Biol* **1043**, 227-256 (2017). https://doi.org:10.1007/978-3-319-70178-3_12
- 15 François, M. *et al.* High-fat diet increases ghrelin-expressing cells in stomach, contributing to obesity. *Nutrition* **32**, 709-715 (2016). <https://doi.org:10.1016/j.nut.2015.12.034>
- 16 Gomez, G., Han, S., Englander, E. W. & Greeley, G. H., Jr. Influence of a long-term high-fat diet on ghrelin secretion and ghrelin-induced food intake in rats. *Regul Pept* **173**, 60-63 (2012). <https://doi.org:10.1016/j.regpep.2011.09.006>
- 17 MacKay, H. *et al.* Rats with a truncated ghrelin receptor (GHSR) do not respond to ghrelin, and show reduced intake of palatable, high-calorie food. *Physiol Behav* **163**, 88-96 (2016). <https://doi.org:10.1016/j.physbeh.2016.04.048>

REVIEWERS' COMMENTS:

Reviewer #1 (Remarks to the Author):

The authors addressed all my questions and concerns adequately. The work is convincing and novel and will be interesting for the wider audience of Communications Biology.

Reviewer #2 (Remarks to the Author):

I thank the Authors for addressing all my comments/suggestions. The manuscript has been substantially improved after revision, and it is my belief that it is now ready for publication in Communications Biology.